# SAMPLING WITH MIRRORED STEIN OPERATORS

**Jiaxin Shi**[1]    **Chang Liu**[2]    **Lester Mackey**[1]
[1] Microsoft Research New England
[2] Microsoft Research Asia
{jiaxinshi,chang.liu,lmackey}@microsoft.com

## ABSTRACT

We introduce a new family of particle evolution samplers suitable for constrained domains and non-Euclidean geometries. Stein Variational Mirror Descent and Mirrored Stein Variational Gradient Descent minimize the Kullback-Leibler (KL) divergence to constrained target distributions by evolving particles in a dual space defined by a mirror map. Stein Variational Natural Gradient exploits non-Euclidean geometry to more efficiently minimize the KL divergence to unconstrained targets. We derive these samplers from a new class of mirrored Stein operators and adaptive kernels developed in this work. We demonstrate that these new samplers yield accurate approximations to distributions on the simplex, deliver valid confidence intervals in post-selection inference, and converge more rapidly than prior methods in large-scale unconstrained posterior inference. Finally, we establish the convergence of our new procedures under verifiable conditions on the target distribution.

## 1 INTRODUCTION

Accurately approximating an unnormalized distribution with a discrete sample is a fundamental challenge in machine learning, probabilistic inference, and Bayesian inference. Particle evolution methods like Stein variational gradient descent (SVGD, Liu & Wang, 2016) tackle this challenge by applying deterministic updates to particles using operators based on Stein's method (Stein, 1972; Gorham & Mackey, 2015; Oates et al., 2017; Liu et al., 2016; Chwialkowski et al., 2016; Gorham & Mackey, 2017) and reproducing kernels (Berlinet & Thomas-Agnan, 2011) to sequentially minimize Kullback-Leibler (KL) divergence. SVGD has found great success in approximating unconstrained distributions for probabilistic learning (Feng et al., 2017; Haarnoja et al., 2017; Kim et al., 2018) but breaks down for constrained targets, like distributions on the simplex (Patterson & Teh, 2013) or the targets of post-selection inference (Taylor & Tibshirani, 2015; Lee et al., 2016; Tian et al., 2016), and fails to exploit informative non-Euclidean geometry (Amari, 1998).

In this work, we derive a family of particle evolution samplers suitable for target distributions with constrained domains and non-Euclidean geometries. Our development draws inspiration from mirror descent (MD) (Nemirovskij & Yudin, 1983), a first-order optimization method that generalizes gradient descent with non-Euclidean geometry. To sample from a distribution with constrained support, our method first maps particles to a dual space. There, we update particle locations using a new class of *mirrored Stein operators* and adaptive reproducing kernels introduced in this work. Finally, the dual particles are mapped back to sample points in the original space, ensuring that all constraints are satisfied. We illustrate this procedure in Fig. 1. In Sec. 3, we develop two algorithms – Mirrored SVGD (MSVGD) and Stein Variational Mirror Descent (SVMD) – with different updates in the dual space; when only a single particle is used, MSVGD reduces to gradient ascent on the log dual space density, and SVMD reduces to mirror ascent on the log target density. In addition, by exploiting the connection between MD and natural gradient descent (Amari, 1998; Raskutti & Mukherjee, 2015), we develop a third algorithm – Stein Variational Natural Gradient (SVNG) – that extends SVMD to unconstrained targets with non-Euclidean geometry.

In Sec. 5, we demonstrate the advantages of our algorithms on benchmark simplex-constrained problems from the literature, constrained sampling problems in post-selection inference (Taylor & Tibshirani, 2015; Lee et al., 2016; Tian et al., 2016), and unconstrained large-scale posterior inference with the Fisher information metric. Finally, we analyze the convergence of our mirrored algorithms in Sec. 6 and discuss our results in Sec. 7.

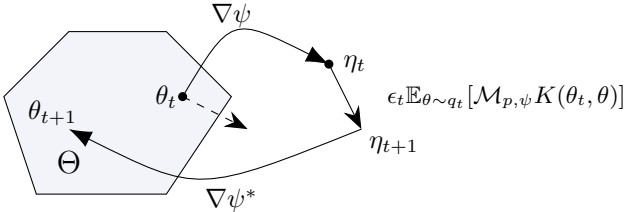

Figure 1: Updating particle approximations in constrained domains $\Theta$. Standard updates like SVGD (dashed arrow) can push particles outside of the support. Our mirrored Stein updates in Alg. 1 (solid arrows) preserve the support by updating particles in a dual space and mapping back to $\Theta$.

**Related work**   Our mirrored Stein operators (6) are instances of diffusion Stein operators in the sense of Gorham & Mackey (2017), but their specific properties have not been studied, nor have they been used to develop sampling algorithms. There is now a large body of work on transferring algorithmic ideas from optimization to MCMC (e.g., Welling & Teh, 2011; Simsekli et al., 2016; Dalalyan, 2017; Durmus et al., 2018; Ma et al., 2019) and SVGD-like sampling methods (e.g., Liu et al., 2019a;b; Zhu et al., 2020; Zhang et al., 2020a). The closest to our work in this space is the recent marriage of mirror descent and MCMC. For example, Hsieh et al. (2018) propose to run Langevin Monte Carlo (LMC, an Euler discretization of the Langevin diffusion) in a mirror space. Zhang et al. (2020b) analyze the convergence properties of the mirror-Langevin diffusion, Chewi et al. (2020) demonstrate its advantages over the Langevin diffusion when using a Newton-type metric, and Ahn & Chewi (2020) study its discretization for MCMC sampling in constrained domains. Relatedly, Patterson & Teh (2013) proposed stochastic Riemannian LMC for sampling on the simplex.

Several modifications of SVGD have been proposed to incorporate geometric information. Riemannian SVGD (RSVGD, Liu & Zhu, 2018) generalizes SVGD to Riemannian manifolds, but, even with the same metric tensor, their updates are more complex than ours: notably they require higher-order kernel derivatives, do not operate in a mirror space, and do not reduce to natural gradient descent when a single particle is used. They also reportedly do not perform well when with scalable stochastic estimates of $\nabla \log p$. Stein Variational Newton (SVN, Detommaso et al., 2018; Chen et al., 2019) introduces second-order information into SVGD. Their algorithm requires an often expensive Hessian computation and need not lead to descent directions, so inexact approximations are employed in practice. Our SVNG can be seen as an instance of matrix SVGD (MatSVGD, Wang et al., 2019) with an adaptive time-dependent kernel discussed in Sec. 4.4, a choice that is not explored in Wang et al. (2019) and which recovers natural gradient descent when $n = 1$ unlike the heuristic kernel constructions of Wang et al. (2019). None of the aforementioned works provide convergence guarantees, and neither SVN nor matrix SVGD deals with constrained domains.

## 2   BACKGROUND: MIRROR DESCENT AND NON-EUCLIDEAN GEOMETRY

Standard gradient descent can be viewed as optimizing a local quadratic approximation to the target function $f$: $\theta_{t+1} = \arg\min_{\theta \in \Theta} \nabla f(\theta_t)^\top \theta + \frac{1}{2\epsilon_t} \|\theta - \theta_t\|_2^2$. When $\Theta \subseteq \mathbb{R}^d$ is constrained, it can be advantageous to replace $\| \cdot \|_2$ with a function $\Psi$ that reflects the geometry of a problem (Nemirovskij & Yudin, 1983; Beck & Teboulle, 2003):

$$\theta_{t+1} = \arg\min_{\theta \in \Theta} \nabla f(\theta_t)^\top \theta + \frac{1}{\epsilon_t} \Psi(\theta, \theta_t). \tag{1}$$

We consider the mirror descent algorithm (Nemirovskij & Yudin, 1983; Beck & Teboulle, 2003) which chooses $\Psi$ to be the Bregman divergence induced by a strongly convex, essentially smooth[1] function $\psi : \Theta \to \mathbb{R} \cup \{\infty\}$: $\Psi(\theta, \theta') = \psi(\theta) - \psi(\theta') - \nabla \psi(\theta')^\top (\theta - \theta')$. When $\Theta$ is a $(d+1)$-simplex $\{\theta : \sum_{i=1}^d \theta_i \leq 1 \text{ and } \theta_i \geq 0 \text{ for } i \in [d]\}$, a common choice of $\psi$ is the negative entropy $\psi(\theta) = \sum_{i=1}^{d+1} \theta_i \log \theta_i$, for $\theta_{d+1} \triangleq 1 - \sum_{i=1}^d \theta_d$. The solution of (1) is given by

$$\theta_{t+1} = \nabla \psi^*(\nabla \psi(\theta_t) - \epsilon_t \nabla f(\theta_t)), \tag{2}$$

where $\psi^*(\eta) \triangleq \sup_{\theta \in \Theta} \eta^\top \theta - \psi(\theta)$ is the convex conjugate of $\psi$ and $\nabla \psi$ is a bijection from $\Theta$ to $\mathrm{dom}(\psi^*)$ with inverse map $(\nabla \psi)^{-1} = \nabla \psi^*$. We can view the update in (2) as first mapping $\theta_t$ to $\eta_t$ by $\nabla \psi$, applying the update $\eta_{t+1} = \eta_t - \epsilon_t \nabla f(\theta_t)$, and mapping back through $\theta_{t+1} = \nabla \psi^*(\eta_{t+1})$.

---

[1]$\psi$ is continuously differentiable on the interior of $\Theta$ with $\|\nabla \psi(\theta_t)\| \to \infty$ whenever $\theta_t \to \theta \in \partial \Theta$.

Mirror descent can also be viewed as a discretization of the continuous-time dynamics $d\eta_t = -\nabla f(\theta_t)dt$, $\theta_t = \nabla\psi^*(\eta_t)$, which is equivalent to the Riemannian gradient flow (see App. A):

$$d\theta_t = -\nabla^2\psi(\theta_t)^{-1}\nabla f(\theta_t)dt, \quad \text{or equivalently,} \quad d\eta_t = -\nabla^2\psi^*(\eta_t)^{-1}\nabla_{\eta_t}f(\nabla\psi^*(\eta_t))dt, \quad (3)$$

where $\nabla^2\psi(\theta)$ and $\nabla^2\psi^*(\eta)$ are Riemannian metric tensors. In information geometry, the discretization of (3) is known as *natural gradient* descent (Amari, 1998). There is considerable theoretical and practical evidence (Martens, 2014) showing that natural gradient works efficiently in learning.

## 3 STEIN'S IDENTITY AND MIRRORED STEIN OPERATORS

Stein's identity (Stein, 1972) is a tool for characterizing a target distribution $P$ using a so-called *Stein operator*. We assume $P$ has a differentiable density $p$ with a closed convex support $\Theta \subseteq \mathbb{R}^d$. A Stein operator $\mathcal{S}_p$ takes as input functions $g$ from a *Stein set* $\mathcal{G}$ and outputs mean-zero functions under $p$:

$$\mathbb{E}_{\theta\sim p}[(\mathcal{S}_p g)(\theta)] = 0, \quad \text{for all } g \in \mathcal{G}. \quad (4)$$

Gorham & Mackey (2015) proposed the *Langevin Stein operator* given by

$$(\mathcal{S}_p g)(\theta) = g(\theta)^\top \nabla \log p(\theta) + \nabla \cdot g(\theta), \quad (5)$$

where $g$ is a vector-valued function and $\nabla \cdot g$ is its divergence. For an *unconstrained* domain with $\mathbb{E}_p[\|\nabla \log p(\theta)\|_2] < \infty$, Stein's identity (4) holds for this operator whenever $g \in C^1$ is bounded and Lipschitz by (Gorham et al., 2019, proof of Prop. 3). However, on constrained domains $\Theta$, Stein's identity fails to hold for many reasonable inputs $g$ if $p$ is non-vanishing or explosive at the boundary.

Motivated by this deficiency and by a desire to exploit non-Euclidean geometry, we propose an alternative *mirrored Stein operator*,

$$(\mathcal{M}_{p,\psi}g)(\theta) = g(\theta)^\top \nabla^2\psi(\theta)^{-1}\nabla \log p(\theta) + \nabla \cdot (\nabla^2\psi(\theta)^{-1}g(\theta)), \quad (6)$$

where $\psi$ is a strongly convex, essentially smooth function as in Sec. 2 with $(\nabla^2\psi)^{-1}$ differentiable and Lipschitz on $\Theta$. We derive this operator from the (infinitesimal) generator of the mirror-Langevin diffusion (19) in App. C. The following result, proved in App. I.1, shows that $\mathcal{M}_{p,\psi}$ generates mean-zero functions under $p$ whenever $\nabla^2\psi^{-1}$ suitably cancels the growth of $p$ at the boundary.

**Proposition 1.** *Suppose that* $\nabla^2\psi(\theta)^{-1}\nabla \log p(\theta)$ *and* $\nabla \cdot \nabla^2\psi(\theta)^{-1}$ *are* $p$-*integrable. If* $\lim_{r\to\infty}\int_{\partial\Theta_r} p(\theta)\|\nabla^2\psi(\theta)^{-1}n_r(\theta)\|_2 d\theta = 0$ *for* $\Theta_r \triangleq \{\theta \in \Theta : \|\theta\|_\infty \leq r\}$ *and* $n_r(\theta)$ *the outward unit normal vector[2] to* $\partial\Theta_r$ *at* $\theta$, *then* $\mathbb{E}_p[(\mathcal{M}_{p,\psi}g)(\theta)] = 0$ *if* $g \in C^1$ *is bounded Lipschitz.*

**Example 1 (Dirichlet $p$, Negative entropy $\psi$).** When $\theta_{1:d+1} \sim \text{Dir}(\alpha)$ for $\alpha \in \mathbb{R}_+^{d+1}$, even setting $g(\theta) = \mathbf{1}$ in (5) need not cause the identity to hold when some $\alpha_j \leq 1$. However, when $\psi(\theta) = \sum_{j=1}^{d+1}\theta_j\log\theta_j$, we show in App. B that the conditions of Prop. 1 are met for any $\alpha$. Remarkably, the mirror-Langevin diffusion for our choice of $\psi$ is the Wright-Fisher diffusion (Ethier, 1976) which Gan et al. (2017) recently used to bound distances to Dirichlet distributions.

## 4 SAMPLING WITH MIRRORED STEIN OPERATORS

Liu & Wang (2016) pioneered the idea of using Stein operators to approximate a target distribution with particles. Their popular SVGD algorithm updates each particle in its approximation by applying the update rule $\theta_{t+1} = \theta_t + \epsilon_t g_t(\theta_t)$ for a chosen mapping $g_t : \mathbb{R}^d \to \mathbb{R}^d$. Specifically, SVGD chooses the mapping $g_t^*$ that leads to the largest decrease in KL divergence to $p$ in the limit as $\epsilon_t \to 0$. The following theorem summarizes their main findings.

**Theorem 2** (Liu & Wang, 2016, Thm. 3.1). *Suppose* $(\theta_t)_{t\geq 0}$ *satisfies* $d\theta_t = g_t(\theta_t)dt$ *for bounded Lipschitz* $g_t \in C^1 : \mathbb{R}^d \to \mathbb{R}^d$ *and that* $\theta_t$ *has density* $q_t$ *with* $\mathbb{E}_{q_t}[\|\nabla \log q_t\|_2] < \infty$. *If* $\text{KL}(q_t\|p) \triangleq \mathbb{E}_{q_t}[\log(q_t/p)]$ *exists then, for the Langevin Stein operator* $\mathcal{S}_p$ (5),

$$\frac{d}{dt}\text{KL}(q_t\|p) = -\mathbb{E}_{\theta_t\sim q_t}[(\mathcal{S}_p g_t)(\theta_t)]. \quad (7)$$

---

[2]For a closed convex set whose boundary $\partial\Theta$ can be locally represented as $F(\theta_1,\ldots,\theta_d) = 0$, its unit normal vector is defined as $n(\theta) = \pm\frac{\nabla F(\theta_1,\ldots,\theta_d)}{\|\nabla F(\theta_1,\ldots,\theta_d)\|_2}$.

---

**Algorithm 1** Mirrored Stein Variational Gradient Descent & Stein Variational Mirror Descent

---

**Input:** density $p$ on $\Theta$, kernel $k$, mirror function $\psi$, particles $(\theta_0^i)_{i=1}^n \subset \Theta$, step sizes $(\epsilon_t)_{t=1}^T$
**Init:** $\eta_0^i \leftarrow \nabla\psi(\theta_0^i)$ for $i \in [n]$
**for** $t = 0 : T$ **do**
    **if** SVMD **then** $K_t \leftarrow K_{\psi,t}$ (13) **else** $K_t \leftarrow kI$ (MSVGD)
    for $i \in [n], \eta_{t+1}^i \leftarrow \eta_t^i + \epsilon_t \frac{1}{n}\sum_{j=1}^n \mathcal{M}_{p,\psi}K_t(\theta_t^i, \theta_t^j)$    (for $\mathcal{M}_{p,\psi}K_t(\cdot, \theta)$ defined in Thm. 3)
    for $i \in [n], \theta_{t+1}^i \leftarrow \nabla\psi^*(\eta_{t+1}^i)$
**return** $\{\theta_{T+1}^i\}_{i=1}^n$.

---

To improve its current particle approximation, SVGD finds the choice of $g_t$ that most quickly decreases $\mathrm{KL}(q_t\|p)$ at time $t$, i.e., it minimizes $\frac{d}{dt}\mathrm{KL}(q_t\|p)$ over a set of candidate directions $g_t$. SVGD finds $g_t$ in a reproducing kernel Hilbert space (RKHS, Berlinet & Thomas-Agnan, 2011) norm ball $\mathcal{B}_{\mathcal{H}^d} = \{g : \|g\|_{\mathcal{H}^d} \leq 1\}$, where $\mathcal{H}^d$ is the product RKHS containing vector-valued functions with each component in the RKHS $\mathcal{H}$ of $k$. Then the optimal $g_t^* \in \mathcal{B}_{\mathcal{H}^d}$ that minimizes (7) is

$$g_t^* \propto g_{q_t,k}^* \triangleq \mathbb{E}_{\theta_t \sim q_t}[k(\theta_t, \cdot)\nabla \log p(\theta_t) + \nabla_{\theta_t} k(\theta_t, \cdot)] = \mathbb{E}_{\theta_t \sim q_t}[\mathcal{S}_p K_k(\cdot, \theta_t)],$$

where we let $K_k(\theta, \theta') = k(\theta, \theta')I$, and $\mathcal{S}_p K_k(\cdot, \theta)$ denotes applying $\mathcal{S}_p$ to each row of $K_k(\cdot, \theta)$. SVGD has found great success in approximating unconstrained target distributions $p$ but breaks down for constrained targets and fails to exploit non-Euclidean geometry. Our goal is to develop new particle evolution samplers suitable for constrained domains and non-Euclidean geometries.

## 4.1 MIRRORED DYNAMICS

SVGD encounters two difficulties when faced with a constrained support. First, the SVGD updates can push the random variable $\theta_t$ outside of its support $\Theta$, rendering all future updates undefined. Second, Stein's identity (4) often fails to hold for candidate directions in $\mathcal{B}_{\mathcal{H}^d}$ (cf. Ex. 1). When this occurs, SVGD need not converge to $p$ as $p$ is not a stationary point of its dynamics (i.e., $\frac{d}{dt}\mathrm{KL}(q_t\|p)|_{q_t=p} \neq 0$ when $q_t = p$). Inspired by mirror descent (Nemirovskij & Yudin, 1983), we consider the following *mirrored* dynamics

$$\theta_t = \nabla\psi^*(\eta_t) \quad \text{for} \quad d\eta_t = g_t(\theta_t)dt, \quad \text{or, equivalently,} \quad d\theta_t = \nabla^2\psi(\theta_t)^{-1}g_t(\theta_t)dt, \quad (8)$$

where $g_t : \Theta \to \mathbb{R}^d$ now represents the update direction in $\eta$ space. The inverse mirror map $\nabla\psi^*$ automatically guarantees that $\theta_t$ belongs to the constrained domain $\Theta$. Since $\psi$ is strongly convex and $\nabla^2\psi^{-1}$ is bounded Lipschitz, from Thm. 2 it follows for any bounded Lipschitz $g_t$ that

$$\frac{d}{dt}\mathrm{KL}(q_t\|p) = -\mathbb{E}_{\theta_t \sim q_t}[(\mathcal{M}_{p,\psi}g_t)(\theta_t)], \quad (9)$$

where $\mathcal{M}_{p,\psi}$ is the mirrored Stein operator (6). In the following sections, we propose three new deterministic sampling algorithms by seeking the optimal direction $g_t$ that minimizes (9) over different function classes. Thm. 3 (proved in App. I.2) forms the basis of our analysis.

**Theorem 3** (Optimal mirror updates in RKHS). *Suppose $(\theta_t)_{t\geq 0}$ follows the mirrored dynamics* (8). *Let $\mathcal{H}_K$ denote the RKHS of a matrix-valued kernel $K : \Theta \times \Theta \to \mathbb{S}^{d\times d}$ (Micchelli & Pontil, 2005). Then, the optimal direction of $g_t$ that minimizes* (9) *in the norm ball $\mathcal{B}_{\mathcal{H}_K} \triangleq \{g : \|g\|_{\mathcal{H}_K} \leq 1\}$ is*

$$g_t^* \propto g_{q_t,K}^* \triangleq \mathbb{E}_{\theta_t \sim q_t}[\mathcal{M}_{p,\psi}K(\cdot, \theta_t)], \quad (10)$$

*where $\mathcal{M}_{p,\psi}K(\cdot, \theta)$ applies $\mathcal{M}_{p,\psi}$* (6) *to each row of the matrix-valued function $K_\theta = K(\cdot, \theta)$.*

## 4.2 MIRRORED STEIN VARIATIONAL GRADIENT DESCENT

Following the pattern of SVGD, one can choose the $K$ of Thm. 3 to be $K_k(\theta, \theta') = k(\theta, \theta')I$, where $k$ is any scalar-valued kernel. In this case, the resulting update $g_{q_t,K_k}^*(\cdot) = \mathbb{E}_{\theta_t \sim q_t}[\mathcal{M}_{p,\psi}K_k(\cdot, \theta_t)]$ is equivalent to running SVGD in the dual $\eta$ space before mapping back to $\Theta$.

**Theorem 4** (Mirrored SVGD updates). *In the setting of Thm. 3, let $k_\psi(\eta, \eta') = k(\nabla\psi^*(\eta), \nabla\psi^*(\eta'))$, $p_H(\eta) = p(\nabla\psi^*(\eta)) \cdot |\det\nabla^2\psi^*(\eta)|$ denote the density of $\eta = \nabla\psi(\theta)$ when $\theta \sim p$, and $q_{t,H}$ denote the distribution of $\eta_t$ under the mirrored dynamics* (8). *If $K_k = kI$,*

$$g_{q_t,K_k}^*(\theta') = \mathbb{E}_{\eta_t \sim q_{t,H}}[k_\psi(\eta_t, \eta')\nabla\log p_H(\eta_t) + \nabla_{\eta_t}k_\psi(\eta_t, \eta')] \quad \forall\theta' \in \Theta, \eta' = \nabla\psi(\theta'). \quad (11)$$

The proof is in App. I.3. By discretizing the dynamics $d\eta_t = g^*_{q_t, K_k}(\theta_t) dt$ and initializing with any particle approximation $q_0 = \frac{1}{n} \sum_{i=1}^n \delta_{\theta_0^i}$, we obtain *Mirrored SVGD (MSVGD)*, our first algorithm for sampling in constrained domains. The details are summarized in Alg. 1.

When only a single particle is used ($n = 1$) and the differentiable input kernel satisfies $\nabla k(\theta, \theta) = 0$, the MSVGD update (11) reduces to gradient descent on $-\log p_H(\eta)$. Note however that the modes of the mirrored density $p_H(\eta)$ need not match those of the target density $p(\theta)$ (see the examples in App. E). Since we are primarily interested in the $\theta$-space density, it is natural to ask whether there exists a mirrored dynamics that reduces to finding the mode of $p(\theta)$ in this limiting case. In the next section, we give an answer to this question by designing an adaptive reproducing kernel that yields a mirror descent-like update.

## 4.3 Stein Variational Mirror Descent

Our second sampling algorithm for constrained problems is called *Stein Variational Mirror Descent (SVMD)*. We start by introducing a new matrix-valued kernel that incorporates the metric $\nabla^2 \psi$ and evolves with the distribution $q_t$.

**Definition 1** (Kernels for SVMD). *Given a continuous scalar-valued kernel $k$, consider the Mercer representation[3] $k(\theta, \theta') = \sum_{i \geq 1} \lambda_{t,i} u_{t,i}(\theta) u_{t,i}(\theta')$ w.r.t. $q_t$, where $u_{t,i}$ is an eigenfunction satisfying*

$$\mathbb{E}_{\theta_t \sim q_t}[k(\theta, \theta_t) u_{t,i}(\theta_t)] = \lambda_{t,i} u_{t,i}(\theta). \tag{12}$$

*For $k_t^{1/2}(\theta, \theta') \triangleq \sum_{i \geq 1} \lambda_{t,i}^{1/2} u_{t,i}(\theta) u_{t,i}(\theta')$, we define the adaptive SVMD kernel at time $t$,*

$$K_{\psi,t}(\theta, \theta') \triangleq \mathbb{E}_{\theta_t \sim q_t}[k_t^{1/2}(\theta, \theta_t) \nabla^2 \psi(\theta_t) k_t^{1/2}(\theta_t, \theta')]. \tag{13}$$

By Thm. 3, the optimal update direction for the SVMD kernel ball is $g^*_{q_t, K_{\psi,t}} = \mathbb{E}_{q_t}[\mathcal{M}_{p,\psi} K_{\psi,t}(\cdot, \theta_t)]$. We obtain the SVMD algorithm (summarized in Alg. 1) by discretizing $d\eta_t = g^*_{q_t, K_{\psi,t}}(\theta_t) dt$ and initializing with $q_0 = \frac{1}{n} \sum_{i=1}^n \delta_{\theta_0^i}$. Because of the discrete representation of $q_t$, $K_{\psi,t}$ takes the form

$$K_{\psi,t}(\theta, \theta') = \sum_{i=1}^n \sum_{j=1}^n \lambda_{t,i}^{1/2} \lambda_{t,j}^{1/2} u_{t,i}(\theta) u_{t,j}(\theta') \Gamma_{t,ij},$$
$$\Gamma_{t,ij} = \frac{1}{n} \sum_{\ell=1}^n u_{t,i}(\theta_t^\ell) u_{t,j}(\theta_t^\ell) \nabla^2 \psi(\theta_t^\ell).$$

Here both $\lambda_{t,j}$ and $u_{t,j}(\theta_t^i)$ can be computed by solving a matrix eigenvalue problem involving the particle set $\{\theta_t^i\}_{i=1}^n$: $B_t v_{t,j} = n \lambda_{t,j} v_{t,j}$, where $B_t = (k(\theta_t^i, \theta_t^j))_{i,j=1}^n \in \mathbb{R}^{n \times n}$ is the Gram matrix of pairwise kernel evaluations at particle locations, and the $i$-th element of $v_{t,j}$ is $u_{t,j}(\theta_t^i)$. To compute $\nabla_\theta K_{\psi,t}(\theta, \theta')$, we differentiate both sides of (12) to find that $\nabla u_{t,j}(\theta) = \frac{1}{\lambda_{t,j}} \sum_{i=1}^n v_{t,j,i} \nabla_\theta k(\theta, \theta_t^i)$. This technique was used in Shi et al. (2018) to estimate gradients of eigenfunctions w.r.t. a continuous $q$. Following their recommendations, we truncate the sum at the $J$-th largest eigenvalues according to a threshold ($\tau \geq \sum_{j=1}^J \lambda_{t,j} / \sum_{j=1}^n \lambda_{t,j}$) to ensure numerical stability.

Notably, SVMD differs from MSVGD only in its choice of kernel, but, whenever $\nabla k(\theta, \theta) = 0$, this change is sufficient to exactly recover mirror descent when $n = 1$.

**Proposition 5** (Single-particle SVMD is mirror descent). *If $n = 1$, then one step of SVMD becomes*

$$\eta_{t+1} = \eta_t + \epsilon_t(k(\theta_t, \theta_t) \nabla \log p(\theta_t) + \nabla k(\theta_t, \theta_t)), \quad \theta_{t+1} = \nabla \psi^*(\eta_{t+1}).$$

## 4.4 Stein Variational Natural Gradient

The fact that SVMD recovers mirror descent as a special case is not only of relevance in constrained problems. We next exploit the connection between MD and natural gradient descent discussed in Sec. 2 to design a new sampler – *Stein Variational Natural Gradient (SVNG)* – that more efficiently approximates unconstrained targets. The idea is to replace the Hessian $\nabla^2 \psi(\cdot)$ in the SVMD dynamics $d\theta_t = \nabla^2 \psi(\theta_t)^{-1} g^*_{q_t, K_{\psi,t}}(\theta_t)$ with a general metric tensor $G(\cdot)$. The result is the Riemannian gradient flow

$$d\theta_t = G(\theta_t)^{-1} g^*_{q_t, K_{G,t}}(\theta_t) dt \quad \text{with} \quad K_{G,t}(\theta, \theta') \triangleq \mathbb{E}_{\theta_t \sim q_t}[k^{1/2}(\theta, \theta_t) G(\theta_t) k^{1/2}(\theta_t, \theta')]. \tag{14}$$

---

[3]See App. F for background on Mercer representations in non-compact domains.

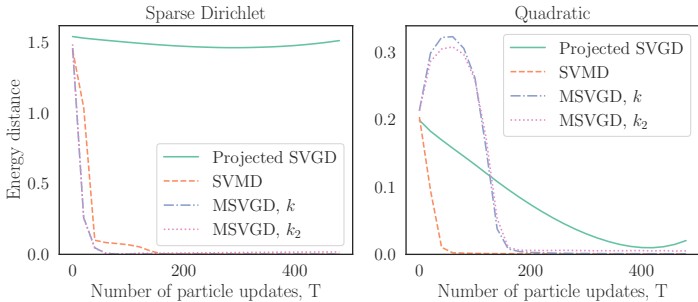

Figure 2: Quality of 50-particle approximations to 20-dimensional distributions on the simplex after $T$ particle updates. (Left) Sparse Dirichlet posterior of Patterson & Teh (2013). (Right) Quadratic simplex target of Ahn & Chewi (2020). Details of the target distributions are in App. G.1.

Given any initial particle approximation $q_0 = \frac{1}{n} \sum_{i=1}^{n} \delta_{\theta_0^i}$, we discretize these dynamics to obtain the unconstrained SVNG sampler of Alg. 2 in the appendix. SVNG can be seen as an instance of MatSVGD (Wang et al., 2019) with a new adaptive time-dependent kernel $G^{-1}(\theta) K_{G,t}(\theta, \theta') G^{-1}(\theta')$. However, similar to Prop. 5 and unlike the heuristic kernels of Wang et al. (2019), SVNG reduces to natural gradient ascent for finding the mode of $p(\theta)$ when $n = 1$. SVNG is well-suited to Bayesian inference problems where the target is a posterior distribution $p(\theta) \propto \pi(\theta)\pi(y|\theta)$. There, the metric tensor $G(\theta)$ can be set to the Fisher information matrix $\mathbb{E}_{\pi(y|\theta)}[\nabla \log \pi(y|\theta) \nabla \log \pi(y|\theta)^\top]$ of the data likelihood $\pi(y|\theta)$. Ample precedent from natural gradient variational inference (Hoffman et al., 2013; Khan & Nielsen, 2018) and Riemannian MCMC (Patterson & Teh, 2013) suggests that encoding problem geometry in this manner often leads to more rapid convergence.

## 5 EXPERIMENTS

We next conduct a series of simulated and real-data experiments to assess (1) distributional approximation on the simplex, (2) frequentist confidence interval construction for (constrained) post-selection inference, and (3) large-scale posterior inference with non-Euclidean geometry. To compare with standard SVGD on constrained domains and to prevent its particles from exiting the domain $\Theta$, we introduce a Euclidean projection onto $\Theta$ following each SVGD update. For SVMD, we need to solve an eigenvalue problem, which costs $O(n^3)$ time. In practice the number of particles used for particle evolution algorithms is relatively small, even for SVGD, due to the $O(n^2)$ cost of updates. We have produced a practical SVMD implementation that is computationally competitive with MSVGD and SVGD for standard particle counts like $n = 50$ (used in all experiments).

### 5.1 APPROXIMATION QUALITY ON THE SIMPLEX

We first measure distributional approximation quality using two 20-dimensional simplex-constrained targets: the sparse Dirichlet posterior of Patterson & Teh (2013) extended to 20 dimensions and the quadratic simplex target of Ahn & Chewi (2020). The Dirichlet target mimics the multimodal sparse conditionals that arise in latent Dirichlet allocation (Blei et al., 2003) but induces a log concave density in $\eta$ space, while the quadratic is log-concave in $\theta$ space. In Fig. 2, we compare the quality of MSVGD, SVMD, and projected SVGD with 50 particles and inverse multiquadric kernel $k$ (Gorham & Mackey, 2017) by computing the energy distance (Székely & Rizzo, 2013) to a surrogate ground truth sample of size 1000 (drawn i.i.d. or, in the quadratic case, from the No-U-Turn Sampler (Hoffman & Gelman, 2014)). We also compare to MSVGD with $k_2(\theta, \theta') = k(\nabla \psi(\theta), \nabla \psi(\theta'))$, a choice which corresponds to running SVGD in the dual space with kernel $k$ by Thm. 4 and which ensures the convergence of MSVGD to $p$ by the upcoming Thms. 6 to 8.

In the quadratic case, SVMD is favored over MSVGD as it is able to exploit the log-concavity of $p(\theta)$. In contrast, for the multimodal sparse Dirichlet with $p(\theta)$ unbounded near the boundary, MSVGD converges slightly more rapidly than SVMD by exploiting the log concave structure in $\eta$ space. This parallels the observation of Hsieh et al. (2018) that LMC in the mirror space outperforms Riemannian LMC for sparse Dirichlet distributions. Projected SVGD fails to converge to the target in both cases and has particular difficulty in approximating the sparse Dirichlet target with unbounded density.

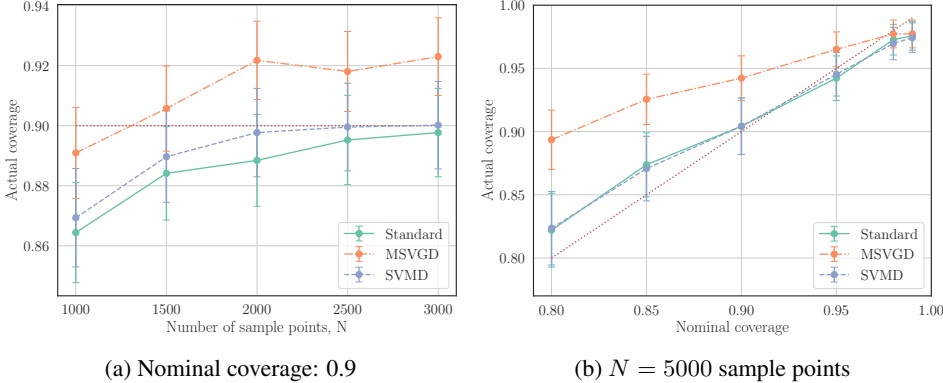

(a) Nominal coverage: 0.9                                 (b) $N = 5000$ sample points

Figure 3: Coverage of post-selection CIs across (a) 500 / (b) 200 replications of simulation of Sepehri & Markovic (2017).

MSVGD with $k$ and $k_2$ perform very similarly, but we observe that $k$ yields better approximation quality upon convergence. Therefore, we employ $k$ in the remaining MSVGD experiments.

## 5.2 CONFIDENCE INTERVALS FOR POST-SELECTION INFERENCE

We next apply our algorithms to the constrained sampling problems that arise in post-selection inference (Taylor & Tibshirani, 2015; Lee et al., 2016). Specifically, we consider the task of forming valid confidence intervals (CIs) for regression parameters selected using the randomized Lasso (Tian et al., 2016) with data $X \in \mathbb{R}^{\tilde{n} \times p}$ and $y \in \mathbb{R}^{\tilde{n}}$ and user-generated randomness $w \in \mathbb{R}^p$ from a log-concave distribution with density $g$. The randomized Lasso returns $\hat{\beta} \in \mathbb{R}^p$ with non-zero coefficients denoted by $\hat{\beta}_E$ and their signs by $s_E$. It is common practice to report least squares CIs for $\beta_E$ by running a linear regression on the selected features $E$. However, since $E$ is chosen based on the same data, the resulting CIs are often invalid.

Post-selection inference solves this problem by conditioning the inference on the knowledge of $E$ and $s_E$. To construct valid CIs, it suffices to approximate the *selective distribution* with support $\{\hat{\beta}_E, u_{-E} : s_E \odot \hat{\beta}_E > 0, \ u_{-E} \in [-1, 1]^{p-|E|}\}$ and density

$$\hat{g}(\hat{\beta}_E, u_{-E}) \propto g\big(X^\top y - \big(\begin{smallmatrix} X_E^\top X_E + \epsilon I_{|E|} \\ X_{-E}^\top X_E \end{smallmatrix}\big)\hat{\beta}_E + \lambda\big(\begin{smallmatrix} s_E \\ u_{-E} \end{smallmatrix}\big)\big).$$

In our experiments, we integrate out $u_{-E}$ analytically, following Tian et al. (2016), and reparameterize $\hat{\beta}_E$ as $s_E \odot |\hat{\beta}_E|$ to obtain a log-concave density of $|\hat{\beta}_E|$ supported on the nonnegative orthant with mirror function $\psi(\theta) = \sum_{j=1}^d (\theta_j \log \theta_j - \theta_j)$. In Fig. 4a we show the example of a 2D selective distribution using samples drawn by NUTS (Hoffman & Gelman, 2014). We also plot the results by projected SVGD, SVMD, and MSVGD in this example. Projected SVGD fails to approximate the target with many samples gathering at the truncation boundary, while the samples by MSVGD and SVMD closely resemble the truth.

We then compare our methods with the standard `norejection` MCMC approach of the `selectiveInference` R package (Tibshirani et al., 2019) using the example simulation setting described in Sepehri & Markovic (2017) and a penalty factor 0.7. To generate $N$ total sample points we run MCMC for $N$ iterations after burn-in or aggregate the particles from $N/n$ independent runs of MSVGD or SVMD with $n = 50$ particles. As $N$ ranges from 1000 to 3000 in Fig. 3a, the MSVGD and SVMD CIs consistently yield higher coverage than the standard 90% CIs. This increased coverage is of particular value for smaller sample sizes, for which the standard CIs tend to undercover. For a much larger sample size of $N = 5000$ in Fig. 3b, the SVMD and standard CIs closely track one another across confidence levels, while MSVGD consistently yields longer CIs with high coverage. The higher coverage of MSVGD is only of value for larger confidence levels at which the other methods begin to undercover.

We next apply our samplers to a post-selection inference task on the HIV-1 drug resistance dataset (Rhee et al., 2006), where we run randomized Lasso (Tian et al., 2016) to find statistically significant mutations associated with drug resistance using susceptibility data on virus isolates.

We take the vitro measurement of log-fold change under the 3TC drug as response and include mutations that had appeared at least 11 times in the dataset as regressors. In Fig. 4b we plot the CIs of selected mutations obtained with $N = 5000$ sample points. We see that the invalid unadjusted least squares CIs can lead to premature conclusions, e.g., declaring mutation 215Y significant when there is insufficient support after conditioning on the selection event. In contrast, mutation 184V, which has known association with drug resistance, is declared significant by all methods even after post-selection adjustment. The MSVGD and SVMD CIs mostly track those of the standard `selectiveInference` method, but their conclusions sometimes differ: e.g., 62Y is flagged as significant by MSVGD and SVMD but not by `selectiveInference`.

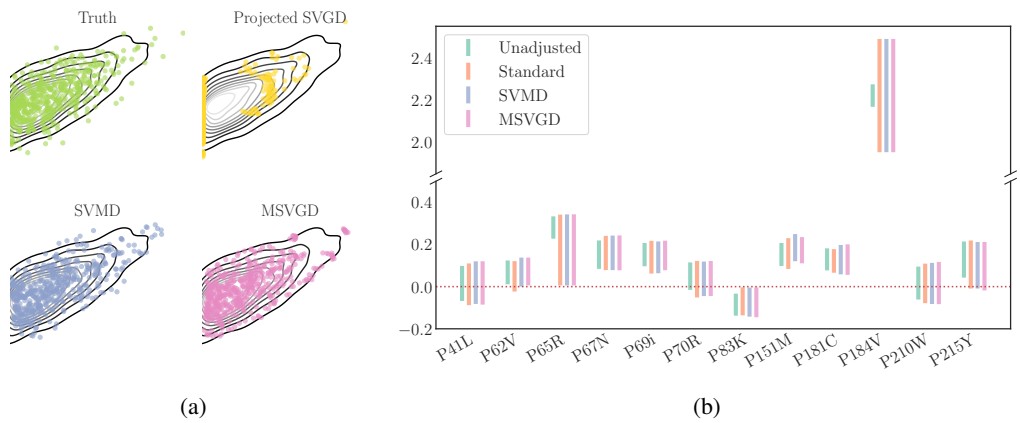

(a)                                                                  (b)

Figure 4: (a) Sampling from a 2D selective density; (b) Unadjusted and post-selection CIs for the mutations selected by the randomized Lasso as candidates for HIV-1 drug resistance (see Sec. 5.2).

### 5.3 LARGE-SCALE POSTERIOR INFERENCE WITH NON-EUCLIDEAN GEOMETRY

Finally, we demonstrate the advantages of exploiting non-Euclidean geometry by recreating the real-data large-scale Bayesian logistic regression experiment of Liu & Wang (2016) with 581,012 datapoints and $d = 54$ feature dimensions. Here, the target $p$ is the posterior distribution over logistic regression parameters. We adopt the Fisher information metric tensor $G$, compare 20-particle SVNG to SVGD and its prior geometry-aware variants RSVGD (Liu & Zhu, 2018) and MatSVGD with average and mixture kernels (Wang et al., 2019), and for all methods use stochastic minibatches of size 256 to scalably approximate each log likelihood query. In Fig. 5, all geometry-aware methods substantially improve the log predictive probability of SVGD. SVNG also strongly outperforms RSVGD and converges to its maximum test probability in half as many steps as MatSVGD (Avg) and more rapidly than MatSVGD (Mixture).

## 6 CONVERGENCE GUARANTEES

We next turn our attention to the convergence properties of our proposed methods. For $K_t$ and $\epsilon_t$ as in Alg. 1, let $(q_t^\infty, q_{t,H}^\infty)$ represent the distributions of the mirrored Stein updates $(\theta_t, \eta_t)$ when $\theta_0 \sim q_0^\infty$ and $\eta_{t+1} = \eta_t + \epsilon_t g_{q_t,K_t}^*(\theta_t)$ for $t \geq 0$. Our first result, proved in App. I.5, shows that if the Alg. 1 initialization $q_{0,H}^n = \frac{1}{n}\sum_{i=1}^n \delta_{\eta_0^i}$ converges in Wasserstein distance to a distribution $q_{0,H}^\infty$ as $n \to \infty$, then, on each round $t > 0$, the output of Alg. 1, $q_t^n = \frac{1}{n}\sum_{i=1}^n \delta_{\theta_t^i}$, converges to $q_t^\infty$.

**Theorem 6** (Convergence of mirrored updates as $n \to \infty$). *Suppose Alg. 1 is initialized with $q_{0,H}^n = \frac{1}{n}\sum_{i=1}^n \delta_{\eta_0^i}$ satisfying $W_1(q_{0,H}^n, q_{0,H}^\infty) \to 0$ for $W_1$ the $L^1$ Wasserstein distance. Define the $\eta$-induced kernel $K_{\nabla\psi^*,t}(\eta, \eta') \triangleq K_t(\nabla\psi^*(\eta), \nabla\psi^*(\eta'))$. If, for some $c_1, c_2 > 0$,*

$$\|\nabla(K_{\nabla\psi^*,t}(\cdot, \eta)\nabla\log p_H(\eta) + \nabla \cdot K_{\nabla\psi^*,t}(\cdot, \eta))\|_{\mathrm{op}} \leq c_1(1 + \|\eta\|_2),$$
$$\|\nabla(K_{\nabla\psi^*,t}(\eta', \cdot)\nabla\log p_H(\cdot) + \nabla \cdot K_{\nabla\psi^*,t}(\eta', \cdot))\|_{\mathrm{op}} \leq c_2(1 + \|\eta'\|_2),$$

*then, $W_1(q_{t,H}^n, q_{t,H}^\infty) \to 0$ and $q_t^n \Rightarrow q_t^\infty$ for each round t.*

Figure 5: Value of non-Euclidean geometry in large-scale Bayesian logistic regression.

**Remark** The pre-conditions hold, for example, whenever $\nabla \log p_H$ is Lipschitz, $\psi$ is strongly convex, and $K_t = kI$ for $k$ bounded with bounded derivatives.

Given a mirrored Stein operator (6), an arbitrary Stein set $\mathcal{G}$, and an arbitrary matrix-valued kernel $K$ we define the *mirrored Stein discrepancy* and *mirrored kernel Stein discrepancy*

$$\mathrm{MSD}(q, p, \mathcal{G}) \triangleq \sup_{g \in \mathcal{G}} \mathbb{E}_q[(\mathcal{M}_{p,\psi} g)(\theta)] \quad \text{and} \quad \mathrm{MKSD}_K(q, p) \triangleq \mathrm{MSD}(q, p, \mathcal{B}_{\mathcal{H}_K}). \quad (15)$$

The former is an example of a diffusion Stein discrepancy (Gorham et al., 2019) and the latter an example of a diffusion kernel Stein discrepancy (Barp et al., 2019). Since the MKSD optimization problem (15) matches that in Thm. 3, we have that $\mathrm{MKSD}_K(q,p) = \|g_{q,K}^*\|_{\mathcal{H}_K}$. Our next result, proved in App. I.6, shows that the infinite-particle mirrored Stein updates reduce the KL divergence to $p$ whenever the step size is sufficiently small and drive MKSD to 0 if, for example, $\epsilon_t = \Omega(\mathrm{MKSD}_{K_t}(q_t^\infty, p)^\alpha)$ for any $\alpha > 0$. We also provide two conditions in App. H that generalize the Stein Log-Sobolev and Stein Poincaré inequalities in Duncan et al. (2019); Korba et al. (2020) and which imply exponential convergence rates of our algorithms in continuous time.

**Theorem 7** (Infinite-particle mirrored Stein updates decrease KL and MKSD). *Assume* $\kappa_1 \triangleq \sup_\theta \|K_t(\theta, \theta)\|_{\mathrm{op}} < \infty$, $\kappa_2 \triangleq \sum_{i=1}^d \sup_\theta \|\nabla_{i,d+i}^2 K_t(\theta, \theta)\|_{\mathrm{op}} < \infty$, $\nabla \log p_H$ *is* $L$-*Lipschitz, and* $\psi$ *is* $\alpha$-*strongly convex. If* $\epsilon_t < 1/(2 \sup_\theta \|\nabla^2 \psi(\theta)^{-1} \nabla g_{q_t^\infty, K_t}^*(\theta) + \nabla g_{q_t^\infty, K_t}^*(\theta)^\top \nabla^2 \psi(\theta)^{-1}\|_{\mathrm{op}})$,

$$\mathrm{KL}(q_{t+1}^\infty \| p) - \mathrm{KL}(q_t^\infty \| p) \le -\left(\epsilon_t - \left(\tfrac{L\kappa_1}{2} + \tfrac{2\kappa_2}{\alpha^2}\right)\epsilon_t^2\right)\mathrm{MKSD}_{K_t}(q_t^\infty, p)^2.$$

Our last result, proved in App. I.7, shows that $q_t^\infty \Rightarrow p$ if $\mathrm{MKSD}_{K_k}(q_t^\infty, p) \to 0$. Hence, by Thms. 6 and 7, $n$-particle MSVGD converges weakly to $p$ if $\epsilon_t$ decays at a suitable rate.

**Theorem 8** ($\mathrm{MKSD}_{K_k}$ determines weak convergence). *Assume* $p_H$ *is distantly dissipative (Eberle, 2016) with* $\nabla \log p_H$ *Lipschitz,* $\psi$ *is strongly convex with continuous* $\nabla \psi^*$, *and* $k(\theta, \theta') = \kappa(\nabla \psi(\theta), \nabla \psi(\theta'))$ *for* $\kappa(x, y) = (c^2 + \|x - y\|_2^2)^\beta$ *with* $\beta \in (-1, 0)$. *Then,* $q_t^\infty \Rightarrow p$ *if* $\mathrm{MKSD}_{K_k}(q_t^\infty, p) \to 0$.

**Remark** The pre-conditions hold, for example, for any Dirichlet target with negative entropy $\psi$.

## 7 DISCUSSION

This paper introduced the mirrored Stein operator along with three new particle evolution algorithms for sampling with constrained domains and non-Euclidean geometries. The first algorithm MSVGD performs SVGD updates in a mirrored space before mapping to the target domain. The other two algorithms are different discretizations of the same continuous dynamics for exploiting non-Euclidean geometry. SVMD is a multi-particle generalization of mirror descent for constrained domains, while SVNG is designed for unconstrained problems with informative metric tensors.

We highlight three limitations. First, like SVGD, our MSVGD require $O(n^2)$ time per update. Second, SVMD and SVNG are more costly than MSVGD due to the adaptive kernel construction. Low-rank kernel approximation may be needed to reduce their complexity. Third, we leave open the question of convergence when stochastic gradient estimates are employed, but we suspect the results of Gorham et al. (2020, Thm. 7) can be extended to our setting. In the future, we hope to deploy our mirrored Stein operators for other inferential tasks on constrained domains including sample quality measurement (Gorham & Mackey, 2015; Huggins & Mackey, 2018), goodness-of-fit testing (Chwialkowski et al., 2016; Liu et al., 2016; Jitkrittum et al., 2017), graphical model inference (Zhuo et al., 2018; Wang et al., 2018), parameter estimation (Barp et al., 2019), thinning (Riabiz et al., 2020), and de novo sampling (Chen et al., 2018; Futami et al., 2019).

REPRODUCIBILITY STATEMENT

See App. G for experimental details and

https://github.com/thjashin/mirror-stein-samplers

for Python and R code replicating all experiments.

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

---

**Algorithm 2** Stein Variational Natural Gradient (SVNG)

---

**Input:** density $p(\theta)$ on $\mathbb{R}^d$, kernel $k$, metric tensor $G(\theta)$, particles $(\theta_0^i)_{i=1}^n$, step sizes $(\epsilon_t)_{t=1}^T$
**for** $t = 0 : T$ **do**
    for $i \in [n], \theta_{t+1}^i \leftarrow \theta_t^i + \epsilon_t G(\theta_t^i)^{-1} g_{G,t}^*(\theta_t^i)$, where
    $g_{G,t}^*(\theta) = \frac{1}{n} \sum_{j=1}^n [K_{G,t}(\theta, \theta_t^j) G(\theta_t^j)^{-1} \nabla \log p(\theta_t^j) + \nabla_{\theta_t^j} \cdot (K_{G,t}(\theta, \theta_t^j) G(\theta_t^j)^{-1})]$ (see (14))
**return** $\{\theta_{T+1}^i\}_{i=1}^n$.

---

## A    MIRROR DESCENT, RIEMANNIAN GRADIENT FLOW, AND NATURAL GRADIENT

The equivalence between the mirror flow $d\eta_t = -\nabla f(\theta_t)dt, \ \theta_t = \nabla \psi^*(\eta_t)dt$ and the Riemannian gradient flow in (3) is a direct result of the chain rule:

$$\frac{d\theta_t}{dt} = -\nabla_{\eta_t}\theta_t \frac{d\eta_t}{dt} = -(\nabla_{\theta_t}\eta_t)^{-1}\frac{d\eta_t}{dt} = -\nabla^2\psi(\theta_t)^{-1}\nabla f(\theta_t), \tag{16}$$

$$\frac{d\eta_t}{dt} = -\nabla f(\theta_t) = -\nabla_{\theta_t}\eta_t \nabla_{\eta_t} f(\nabla\psi^*(\eta_t)) = -\nabla^2\psi^*(\eta_t)^{-1}\nabla_{\eta_t} f(\nabla\psi^*(\eta_t)). \tag{17}$$

Depending on discretizing (16) or (17), there are two natural gradient descent (NGD) updates that can arise from the same gradient flow:

$$\text{NGD (a):} \quad \theta_{t+1} = \theta_t - \epsilon_t \nabla^2\psi(\theta_t)^{-1}\nabla f(\theta_t),$$

$$\text{NGD (b):} \quad \eta_{t+1} = \eta_t - \epsilon_t \nabla^2\psi^*(\eta_t)^{-1}\nabla_{\eta_t} f(\nabla\psi^*(\eta_t)).$$

With finite step sizes $\epsilon_t$, their updates need not be the same and can lead to different optimization paths. Since $\nabla f(\theta_t) = \nabla^2\psi^*(\eta_t)^{-1}\nabla_{\eta_t} f(\nabla\psi^*(\eta_t))$, NGD (b) is equivalent to the dual-space update by mirror descent. This relationship was pointed out in Raskutti & Mukherjee (2015) and has been used for developing natural gradient variational inference algorithms (Khan & Nielsen, 2018). We emphasize, however, our SVNG algorithm developed in Sec. 4.4 corresponds to the discretization in the primal space as in NGD (a). Therefore, it does not require an explicit dual space, and allows replacing $\nabla^2\psi$ with more general information metric tensors.

## B    DETAILS OF EXAMPLE 1

For the entropic mirror map $\psi(\theta) = \sum_{j=1}^{d+1} \theta_j \log\theta_j$, we have $\nabla^2\psi(\theta)^{-1} = \text{diag}(\theta) - \theta\theta^\top$. Note here $\theta$ denotes a $d$-dimensional vector and does not include $\theta_{d+1} = 1 - \sum_{j=1}^d \theta_d$. Since $\Theta$ is a $(d+1)$-simplex, $\partial\Theta$ is composed of $d+1$ faces with $\theta$ in the $j$-th face satisfies $\theta_j = 0$. The outward unit normal vector $n(\theta)$ for the first $d$ faces are $-e_j$ for $1 \le j \le d$, where $e_j$ denotes the $j$-th standard basis of $\mathbb{R}^d$. The outward unit normal vector for the $(d+1)$-st face is a vector with $1/\sqrt{d}$ in all coordinates. Therefore, we have

$$\int_{\partial\Theta} p(\theta)g(\theta)^\top \nabla^2\psi(\theta)^{-1}n(\theta)d\theta = \int_{\partial\Theta} p(\theta)g(\theta)^\top (\text{diag}(\theta) - \theta\theta^\top)n(\theta)d\theta$$

$$= \int_{\partial\Theta} p(\theta)(\theta \odot g(\theta) - \theta\theta^\top g(\theta))^\top n(\theta)d\theta$$

$$= \sum_{j=1}^d \int_{\theta_j=0} p(\theta)(\theta^\top g(\theta) - g_j(\theta))\theta_j d\theta_{-j}$$

$$+ \frac{1}{\sqrt{d}} \int_{\theta_{d+1}=0} p(\theta)\theta^\top g(\theta)\theta_{d+1}d\theta$$

$$= 0,$$

where in the second to last identity we used $\theta^\top \mathbf{1} = 1 - \theta_{d+1}$. Finally, we can verify the condition in Prop. 1 as

$$\lim_{r \to \infty} \int_{\partial\Theta_r} p(\theta)\|\nabla^2\psi(\theta)^{-1}n_r(\theta)\|_2 d\theta = \sup_{\|g\|_\infty \le 1} \int_{\partial\Theta} p(\theta)g(\theta)^\top \nabla^2\psi(\theta)^{-1}n(\theta)d\theta = 0.$$

## C DERIVATION OF THE MIRRORED STEIN OPERATOR

We first review the (overdamped) Langevin diffusion – a Markov process that underlies many recent advances in Stein's method – along with its recent mirrored generalization. The Langevin diffusion with equilibrium density $p$ on $\mathbb{R}^d$ is a Markov process $(\theta_t)_{t \geq 0} \subset \mathbb{R}^d$ satisfying the stochastic differential equation (SDE)

$$d\theta_t = \nabla \log p(\theta_t) dt + \sqrt{2} dB_t \qquad (18)$$

with $(B_t)_{t \geq 0}$ a standard Brownian motion (Bhattacharya & Waymire, 2009, Sec. 4.5).

To identify Stein operators that satisfy (4) for broad classes of targets $p$, Gorham & Mackey (2015) proposed to build upon the generator method of Barbour (1988): First, identify a Markov process $(\theta_t)_{t \geq 0}$ that has $p$ as the equilibrium density; they chose the Langevin diffusion of (18). Next, build a Stein operator based on the (infinitesimal) generator $A$ of the process (Øksendal, 2003, Def. 7.3.1):

$$(Af)(\theta) = \lim_{t \to 0} \tfrac{1}{t}(\mathbb{E}f(\theta_t) - \mathbb{E}f(\theta_0)) \quad \text{for } f : \mathbb{R}^d \to \mathbb{R},$$

as the generator satisfies $\mathbb{E}_{\theta \sim p}[(Af)(\theta)] = 0$ under relatively mild conditions. We use the following theorem to derive the generator of the processes described by SDEs like (18):

**Theorem 9** (Generator of Itô diffusion; Øksendal, 2003, Thm 7.3.3)**.** *Let $(x_t)_{t \geq 0}$ be the Itô diffusion in $\mathcal{X} \subseteq \mathbb{R}^d$ satisfying $dx_t = b(x_t)dt + \sigma(x_t)dB_t$. For any $f \in \overline{C_c^2(\mathcal{X})}$, the (infinitesimal) generator $A$ of $(x_t)_{t \geq 0}$ is*

$$(Af)(x) = b(x)^\top \nabla f(x) + \tfrac{1}{2} \operatorname{Tr}(\sigma(x)\sigma(x)^\top \nabla^2 f(x)).$$

For the Langevin diffusion (18), substituting $\nabla \log p(\cdot)$ for $b(\cdot)$ and $\sqrt{2}I$ for $\sigma(\cdot)$ in Thm. 9, we obtain $Af = (\nabla \log p)^\top \nabla f + \nabla \cdot \nabla f$. Replacing $\nabla f$ with a vector-valued function $g$ gives the Langevin Stein operator in (5).

To derive a Stein operator that works well for constrained domains, we consider the Riemannian Langevin diffusion (Patterson & Teh, 2013; Xifara et al., 2014; Ma et al., 2015) that extends the Langevin diffusion to non-Euclidean geometries encoded in a positive definite *metric tensor* $G(\theta)$:

$$d\theta_t = (G(\theta_t)^{-1} \nabla \log p(\theta_t) + \nabla \cdot G(\theta_t)^{-1})dt + \sqrt{2}G(\theta_t)^{-1/2}dB_t.^4$$

We show in App. D that the choice $G = \nabla^2 \psi$ yields the recent mirror-Langevin diffusion (Zhang et al., 2020b; Chewi et al., 2020)

$$\theta_t = \nabla \psi^*(\eta_t), \quad d\eta_t = \nabla \log p(\theta_t)dt + \sqrt{2}\nabla^2 \psi(\theta_t)^{1/2}dB_t. \qquad (19)$$

According to Thm. 9, the generator of the mirror-Langevin diffusion described by (20) is

$$\begin{aligned} (A_{p,\psi}f)(\theta) &= (\nabla^2 \psi(\theta)^{-1} \nabla \log p(\theta) + \nabla \cdot \nabla^2 \psi(\theta)^{-1})^\top \nabla f(\theta) + \operatorname{Tr}(\nabla^2 \psi(\theta)^{-1} \nabla^2 f(\theta)) \\ &= \nabla f(\theta)^\top \nabla^2 \psi(\theta)^{-1} \nabla \log p(\theta) + \nabla \cdot (\nabla^2 \psi(\theta)^{-1} \nabla f(\theta)). \end{aligned}$$

Now substituting $g(\theta)$ for $\nabla f(\theta)$, we obtain the associated mirrored Stein operator:

$$(\mathcal{M}_{p,\psi}g)(\theta) = g(\theta)^\top \nabla^2 \psi(\theta)^{-1} \nabla \log p(\theta) + \nabla \cdot (\nabla^2 \psi(\theta)^{-1} g(\theta)).$$

## D RIEMANNIAN LANGEVIN DIFFUSIONS AND MIRROR-LANGEVIN DIFFUSIONS

Zhang et al. (2020b) pointed out (19) is a particular case of the Riemannian LD. However, they did not give an explicit derivation. The Riemannian LD (Patterson & Teh, 2013; Xifara et al., 2014; Ma et al., 2015) with $\nabla^2 \psi(\cdot)$ as the metric tensor is

$$d\theta_t = (\nabla^2 \psi(\theta_t)^{-1} \nabla \log p(\theta_t) + \nabla \cdot \nabla^2 \psi(\theta_t)^{-1})dt + \sqrt{2}\nabla^2 \psi(\theta_t)^{-1/2}dB_t. \qquad (20)$$

To see the connection with mirror-Langevin diffusion, we would like to obtain the SDE that describes the evolution of $\eta_t = \nabla \psi(\theta_t)$ under the diffusion. This requires the following theorem that provides the analog of the "chain rule" in SDEs.

---

[4] A matrix divergence $\nabla \cdot G(\theta)$ is the vector obtained by computing the divergence of each row of $G(\theta)$.

**Theorem 10** (Itô formula; Øksendal, 2003, Thm 4.2.1). *Let $(x_t)_{t\geq 0}$ be an Itô process in $\mathcal{X} \subset \mathbb{R}^d$ satisfying $dx_t = b(x_t)dt + \sigma(x_t)dB_t$. Let $f(x) \in C^2 : \mathbb{R}^d \to \mathbb{R}^{d'}$. Then $y_t = f(x_t)$ is again an Itô process, and its $i$-th dimension satisfies*

$$dy_{t,i} = (\nabla f_i(x_t)^\top b(x_t) + \frac{1}{2}\operatorname{Tr}(\nabla^2 f_i(x_t)\sigma(x_t)\sigma(x_t)^\top)dt + \nabla f_i(x_t)^\top\sigma(x_t)dB_t.$$

Substituting $\nabla\psi$ for $f$ in Thm. 10, we have the SDE of $\eta_t = \nabla\psi(\theta_t)$ as

$$d\eta_t = (\nabla\log p(\theta_t) + \nabla^2\psi(\theta_t)\nabla\cdot\nabla^2\psi(\theta_t)^{-1} + h(\theta_t))dt + \sqrt{2}\nabla^2\psi(\theta_t)^{1/2}dB_t,$$

where $h(\theta_t)_i = \operatorname{Tr}(\nabla^2_{\theta_t}(\nabla_{\theta_{t,i}}\psi(\theta_t))\nabla^2\psi(\theta_t)^{-1})$. Moreover, we have

$$[\nabla^2\psi(\theta_t)\nabla\cdot\nabla^2\psi(\theta_t)^{-1}]_i + \operatorname{Tr}(\nabla^2_{\theta_t}(\nabla_{\theta_{t,i}}\psi(\theta_t))\nabla^2\psi(\theta_t)^{-1})$$

$$= \sum_{\ell=1}^d\sum_{j=1}^d \nabla^2\psi(\theta_t)_{ij}\nabla_{\theta_{t,\ell}}[\nabla^2\psi(\theta_t)^{-1}]_{j\ell} + \sum_{\ell=1}^d\sum_{j=1}^d \nabla_{\theta_{t,\ell}}\nabla^2\psi(\theta_t)_{ij}[\nabla^2\psi(\theta_t)^{-1}]_{j\ell}$$

$$= \sum_{\ell=1}^d \nabla_{\theta_{t,\ell}}\left(\sum_{j=1}^d \nabla^2\psi(\theta_t)_{ij}[\nabla^2\psi(\theta_t)^{-1}]_{j\ell}\right) = \sum_{\ell=1}^d \nabla_{\theta_{t,\ell}}I_{i\ell} = 0.$$

Therefore, the $\eta_t$ diffusion is described by the SDE:

$$d\eta_t = \nabla\log p(\theta_t)dt + \sqrt{2}\nabla^2\psi(\theta_t)^{1/2}dB_t, \quad \theta_t = \nabla\psi^*(\eta_t).$$

## E    MODE MISMATCH UNDER TRANSFORMATIONS

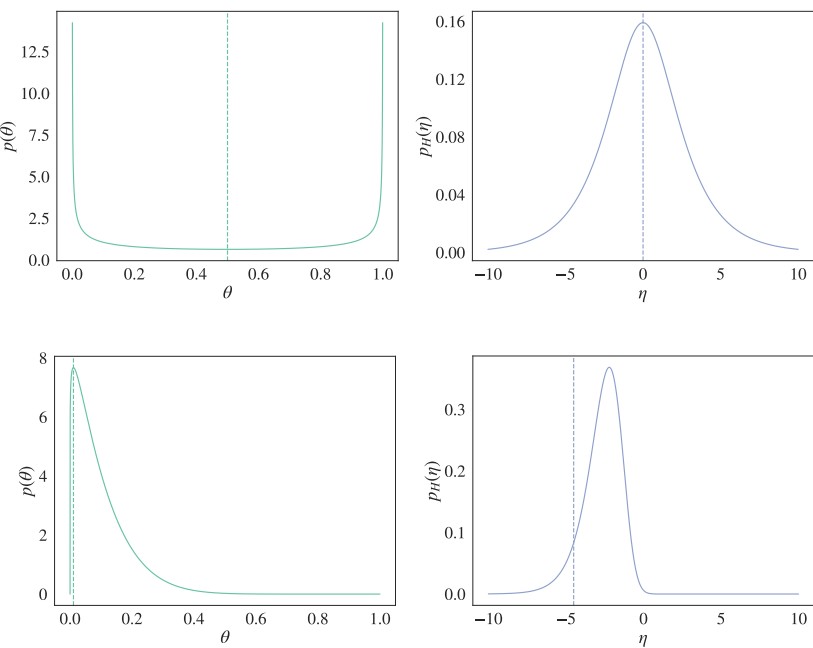

Figure 6: The density functions of the same distribution in $\theta$ (left) and $\eta$ (right) space under the transformation $\eta = \nabla\psi(\theta)$. Each $\theta$ follows a Beta distributions on $[0, 1]$. We choose the negative entropy $\psi(\theta) = \theta\log\theta + (1-\theta)\log(1-\theta)$. Then, the transformation is the logit function $\eta = \log(\theta/(1-\theta))$ and its reverse is the sigmoid function $\theta = 1/(1+e^{-\eta})$. *Top*: $\theta \sim \operatorname{Beta}(0.5, 0.5)$. Dashed lines mark the mode of the transformed density $p_H(\eta)$ and the corresponding $\theta$, which gives the lowest value of $p(\theta)$; *Bottom*: $\theta \sim \operatorname{Beta}(1.1, 10)$. Dashed lines mark the mode of the target density $p(\theta)$ and the corresponding $\eta$, which clearly does not match the mode of $p_H(\eta)$.

## F   Background on Reproducing Kernel Hilbert Spaces

Let $\mathcal{H}$ be a Hilbert space of functions defined on $\mathcal{X}$ and taking their values in $\mathbb{R}$. We say $k$ is a reproducing kernel (or kernel) of $\mathcal{H}$ if $\forall x \in \mathcal{X}, k(x, \cdot) \in \mathcal{H}$ and $\forall f \in \mathcal{H}, \langle f, k(x, \cdot)\rangle_\mathcal{H} = f(x)$. $\mathcal{H}$ is called a reproducing kernel Hilbert space (RKHS) if it has a kernel. Kernels are positive definite (p.d.) functions, which means that matrices with the form $(k(x_i, x_j))_{ij}$ are positive semidefinite. For any p.d. function $k$, there is a unique RKHS with $k$ as the reproducing kernel, which can be constructed by the completion of $\{\sum_{i=1}^n a_i k(x_i, \cdot), x_i \in \mathcal{X}, a_i \in \mathbb{R}, i \in \mathbb{N}\}$.

Now we assume $\mathcal{X}$ is a metric space, $k$ is a bounded continuous kernel with the RKHS $\mathcal{H}$, and $\nu$ is a positive measure on $\mathcal{X}$. $L^2(\nu)$ denote the space of all square-integrable functions w.r.t. $\nu$. Then the kernel integral operator $T_k : L^2(\nu) \to L^2(\nu)$ defined by

$$T_k g = \int_\mathcal{X} g(x)k(x, \cdot)d\nu$$

is compact and self-adjoint. Therefore, according to the spectral theorem, there exists an at most countable set of positive eigenvalues $\{\lambda_j\}_{j\in J} \subset \mathbb{R}$ with $\lambda_1 \geq \lambda_2 \geq \ldots$ converging to zero and orthonormal eigenfunctions $\{u_j\}_{j\in J}$ such that

$$T_k u_j = \lambda_j u_j,$$

and $k$ has the representation $k(x, x') = \sum_{j\in J} \lambda_j u_j(x)u_j(x')$ (Mercer's theorem on non-compact domains), where the convergence of the sum is absolute and uniform on compact subsets of $\mathcal{X} \times \mathcal{X}$ (Ferreira & Menegatto, 2009).

## G   Supplementary Experimental Details and Additional Results

In this section, we report supplementary details and additional results from the experiments of Sec. 5. In Secs. 5.1 and 5.2, we use the inverse multiquadric input kernel $k(\theta, \theta') = (1 + \|\theta - \theta'\|_2^2/\ell^2)^{-1/2}$ due to its convergence control properties (Gorham & Mackey, 2017). In the unconstrained experiments of Sec. 5.3, we use the Gaussian kernel $k(\theta, \theta') = \exp(-\|\theta - \theta'\|_2^2/\ell^2)$ for consistency with past work. The bandwidth $\ell$ is determined by the median heuristic (Garreau et al., 2017). We select $\tau$ from $\{0.98, 0.99\}$ for all SVMD experiments. For unconstrained targets, we report, for each method, results from the best fixed step size $\epsilon \in \{0.01, 0.05, 0.1, 0.5, 1\}$ selected on a separate validation set. For constrained targets, we select step sizes adaptively to accommodate rapid density growth near the boundary; specifically, we use RMSProp (Hinton et al., 2012), an extension of the AdaGrad algorithm (Duchi et al., 2011) used in Liu & Wang (2016), and report performance with the best learning rate. Results were recorded on an Intel(R) Xeon(R) CPU E5-2690 v4 @ 2.60GHz and an NVIDIA Tesla P100 PCIe 16GB.

### G.1   Approximation quality on the simplex

The sparse Dirichlet posterior of Patterson & Teh (2013) extended to 20 dimensions features a sparse, symmetric Dir($\alpha$) prior with $\alpha_k = 0.1$ for $k \in \{1, \ldots, 20\}$ and sparse count data $n_1 = 90$, $n_2 = n_3 = 5$, $n_j = 0$ $(j > 3)$, modeled via a multinomial likelihood. The quadratic target satisfies $\log p(\theta) = -\frac{1}{2\sigma^2}\theta^\top A\theta + \text{const}$, where we slightly modify the target density of Ahn & Chewi (2020) to make it less flat by introducing a scale parameter $\sigma = 0.01$. $A \in \mathbb{R}^{19\times19}$ is a positive definite matrix generated by normalizing products of random matrices with i.i.d. elements drawn from Unif$[-1, 1]$.

We initialize all methods with i.i.d samples from Dirichlet(5) to prevent any of the initial particles being too close to the boundary. For each method and each learning rate we apply 500 particle updates. For SVMD we set $\tau = 0.98$. We search the base learning rates of RMSProp in $\{0.1, 0.01, 0.001\}$ for SVMD and MSVGD. Since projected SVGD applies updates in the $\theta$ space, the appropriate learning rate range is smaller than those of SVMD and MSVGD. There we search the base learning rate of RMSProp in $\{0.01, 0.001, 0.0001\}$. For all methods the results under each base learning rate are plotted in Fig. 7.

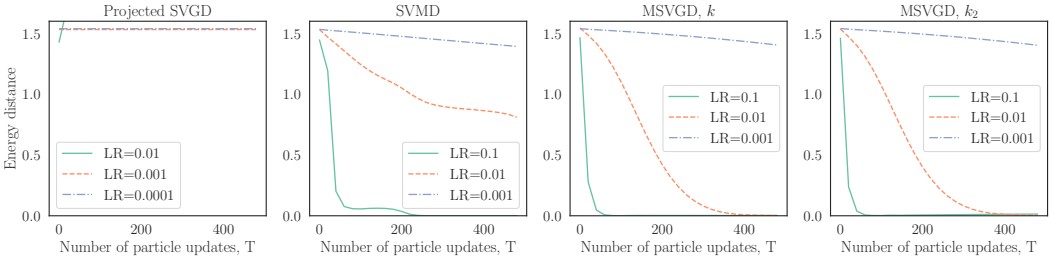

Figure 7: Sampling from a Dirichlet target on a 20-simplex. We plot the energy distance to a ground truth sample of size 1000.

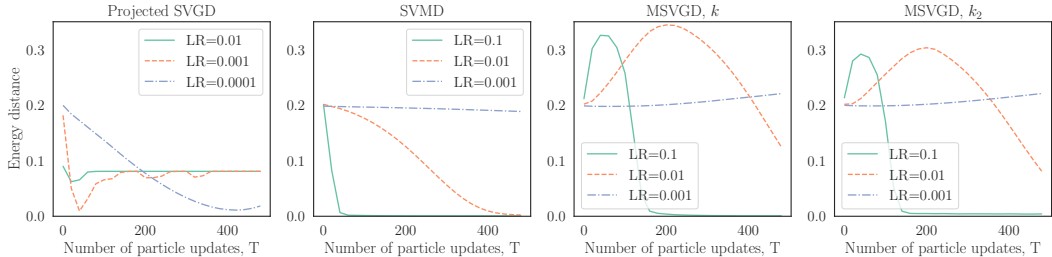

Figure 8: Sampling from a quadratic target on a 20-simplex. We plot the energy distance to a ground truth sample of size 1000 drawn by NUTS (Hoffman & Gelman, 2014).

### G.2 Confidence intervals for post-selection inference

Given a dataset $X \in \mathbb{R}^{\tilde{n} \times p}$, $y \in \mathbb{R}^{\tilde{n}}$, the randomized Lasso (Tian et al., 2016) solves the following problem:

$$\operatorname{argmin}_{\beta \in \mathbb{R}^p} \tfrac{1}{2}\|y - X\beta\|_2^2 + \lambda\|\beta\|_1 - w^\top\beta + \tfrac{\epsilon}{2}\|\beta\|_2^2, \quad w \sim \mathbb{G}.$$

where $\mathbb{G}$ is a user-specified log-concave distribution with density $g$. We choose $\mathbb{G}$ to be zero-mean independent Gaussian distributions while leaving its scale and the ridge parameter $\epsilon$ to be automatically determined by the `randomizedLasso` function of the `selectiveInference` package. We initialize the particles of our SVMD and MSVGD in the following way: First, we map the solution $\hat{\beta}_E$ to the dual space by $\nabla\psi$. Next, we add i.i.d. standard Gaussian noise to $n$ copies of the image in the dual space. Finally, we map the $n$ particles back to the primal space by $\nabla\psi^*$ and use them as the initial locations. Below we discuss the remaining settings and additional results of the simulation and the HIV-1 drug resistance experiment separately.

**Simulation** In our simulation we mostly follow the settings of Sepehri & Markovic (2017) except using a different penalty level $\lambda$ recommended in the `selectiveInference` R package. We set $\tilde{n} = 100$ and $p = 40$. The design matrix $X$ is generated from an equi-correlated model, i.e., each datapoint $x_i \in \mathbb{R}^p$ is generated i.i.d. from $\mathcal{N}(0, \Sigma)$ with $\Sigma_{ii} = 1, \Sigma_{ij} = 0.3$ $(i \neq j)$ and then normalized to have almost unit length. The normalization is done by first centering each dimension by subtracting the mean and dividing the standard deviation of that column of $X$, then additionally multiplying $1/\tilde{n}^{1/2}$. $y$ is generated from a standard Gaussian which is independent of $X$, i.e., we assume the global null setting where the true value of $\beta$ is zero. We set $\lambda$ to be the value returned by `theoretical.lambda` of the `selectiveInference` R package multiplied a coefficient $0.7\tilde{n}$, where the 0.7 adjustment is introduced in the test examples of the R package to reduce the regularization effect so that we have a reasonably large set of selected features when $p = 40$. The base learning rates for SVMD and MSVGD are set to $0.01$ and we run them for $T = 1000$ particle updates. $\tau$ is set to $0.98$ for SVMD.

Our 2D example in Fig. 4a is grabbed from one run of the simulation where there happen to be only 2 features selected by the randomized Lasso. The selective distribution in this case has log-density $\log p(\theta) = -8.07193((2.39859\theta_1 + 1.90816\theta_2 + 2.39751)^2 + (1.18099\theta_2 - 1.46104)^2) + \text{const}, \theta_{1,2} \geq 0$.

The error bars for actual coverage levels in Fig. 3a and Fig. 3b are 95% Wilson intervals (Wilson, 1927), which is known to be more accurate than $\pm 2$ standard deviation intervals for binomial proportions like the coverage. In Fig. 9a and Fig. 9b we additionally plot the average length of the confidence intervals w.r.t. different sample size $N$ and nominal coverage levels. For all three methods the CI widths are very close, although MSVGD consistently has wider intervals than SVMD and `selectiveInference`. This indicates that SVMD can be preferred over MSVGD when both methods produce coverage above the nominal level.

**HIV-1 drug resistance**  We take the vitro measurement of log-fold change under the 3TC drug as response and include mutations that had appeared 11 times in the dataset as regressors. This results in $\tilde{n} = 663$ datapoints with $p = 91$ features. We choose $\lambda$ to be the value returned by `theoretical.lambda` of the `selectiveInference` R package multiplied by $\tilde{n}$. The base learning rates for SVMD and MSVGD are set to $0.01$ and we run them for $T = 2000$ particle updates. $\tau$ is set to $0.99$ for SVMD.

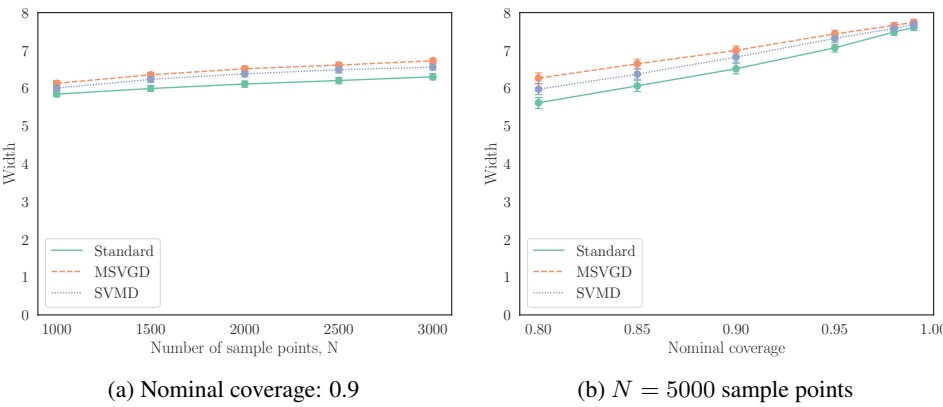

(a) Nominal coverage: 0.9 $\qquad\qquad$ (b) $N = 5000$ sample points

Figure 9: Width of post-selection CIs across (a) 500 / (b) 200 replications of simulation of Sepehri & Markovic (2017).

### G.3 LARGE-SCALE POSTERIOR INFERENCE WITH NON-EUCLIDEAN GEOMETRY

The Bayesian logistic regression model we consider is $\prod_{i=1}^{\tilde{n}} p(y_i|x_i, w)p(w)$, where $p(w) = \mathcal{N}(w|0, I)$, $p(y_i|x_i, w) = \text{Bernoulli}(\sigma(w^\top x_i))$. The bias parameter is absorbed into into $w$ by adding an additional feature 1 to each $x_i$. The gradient of the log density of the posterior distribution of $w$ is $\nabla_w \log p(w|\{y_i, x_i\}_{i=1}^N) = \sum_{i=1}^N x_i(y_i - \sigma(w^\top x_i)) - w$. We choose the metric tensor $\nabla^2 \psi(w)$ to be the Fisher information matrix (FIM) of the likelihood:

$$F = \frac{1}{\tilde{n}} \sum_{i=1}^{\tilde{n}} \mathbb{E}_{p(y_i|w,x_i)}[\nabla_w \log p(y_i|x_i, w)\nabla_w \log p(y_i|x_i, w)^\top]$$

$$= \frac{1}{\tilde{n}} \sum_{i=1}^{\tilde{n}} \sigma(w^\top x_i)(1 - \sigma(w^\top x_i))x_i x_i^\top.$$

Following Wang et al. (2019), for each iteration $r$ ($r \geq 1$), we estimate the sum with a stochastic minibatch $\mathcal{B}_r$ of size 256: $\hat{F}_{\mathcal{B}_r} = \frac{\tilde{n}}{|\mathcal{B}_r|} \sum_{i \in \mathcal{B}_r} \sigma(w^\top x_i)(1 - \sigma(w^\top x_i))x_i x_i^\top$ and approximate the FIM with a moving average across iterations:

$$\hat{F}_r = \rho_r \hat{F}_{r-1} + (1 - \rho_r)\hat{F}_{\mathcal{B}_r}, \quad \text{where } \rho_r = \min(1 - 1/r, 0.95).$$

To ensure the positive definiteness of the FIM, a damping term $0.01I$ is added before taking the inverse. For RSVGD and SVNG, the gradient of the inverse of FIM is estimated with $\nabla_{w_j} F^{-1} \approx -\hat{F}_r^{-1}(\hat{\nabla}_{w_j}^r F)\hat{F}_r^{-1}$, where $\hat{\nabla}_{w_j}^r F = \rho_r \hat{\nabla}_{w_j}^{r-1} F + (1 - \rho_r)\nabla_{w_j}\hat{F}_{\mathcal{B}_r}$.

We run each method for $T = 3000$ particle updates with learning rates in $\{0.01, 0.05, 0.1, 0.5, 1\}$ and average the results for 5 random trials. $\tau$ is set to $0.98$ for SVNG. For each run, we randomly

keep 20% of the dataset as test data, 20% of the remaining points as the validation set, and all the rest as the training set. The results of each method on validation sets with all choices of learning rates are plotted in Fig. 10. We see that the SVNG updates are very robust to the change in learning rates and is able to accommodate very large learning rates (up to 1) without a significant loss in performance. The results in Fig. 5 are reported with the learning rate that performs best on the validation set.

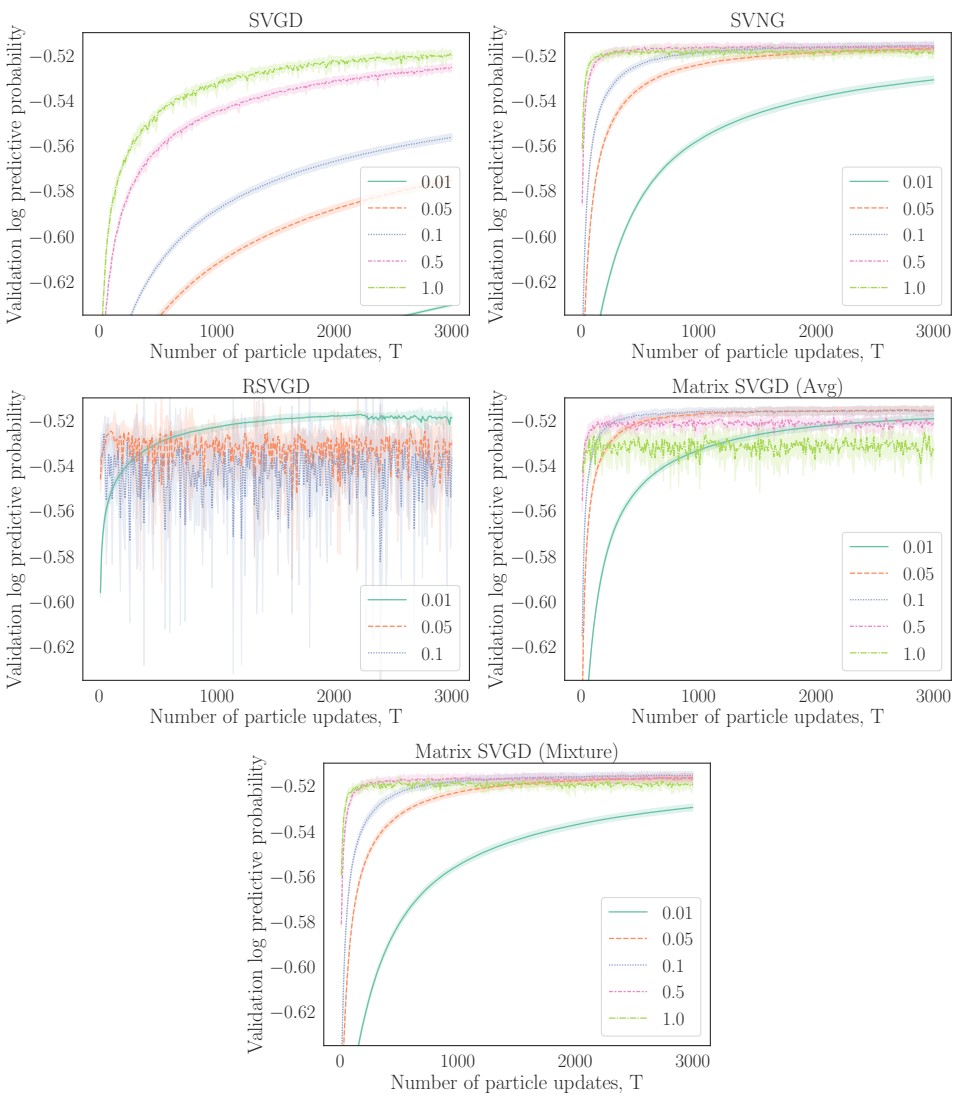

Figure 10: Logistic regression results on validation sets with learning rates in {0.01, 0.05, 0.1, 0.5, 1}. Running RSVGD with learning rates 0.5 and 1 produces numerical errors. Therefore, we did not include them in the plot.

## H  EXPONENTIAL CONVERGENCE OF CONTINUOUS-TIME ALGORITHMS

We derive a time-inhomogeneous generalization of the Stein Log-Sobolev inequality of Duncan et al. (2019) and Korba et al. (2020) which ensures the exponential convergence of our continuous-time algorithms and time-inhomogeneous generalization of the Stein Poincaré inequality of Duncan et al. (2019) which guarantees exponential convergence near equilibrium (i.e., when $q_t$ is sufficiently close to $p$). As the results hold for a generic sequence of kernels $(K_t)_{t \geq 0}$, the implications apply to both MSVGD and SVMD.

**Definition 2** (Mirror Stein Log-Sobolev inequality). *We define the Mirror Stein Log-Sobolev inequality (cf., Korba et al., 2020, Def. 2) as*

$$\mathrm{KL}(q_t\|p) \leq \frac{1}{2\lambda}\mathrm{MKSD}_{K_t}^2(q_t, p) = \frac{1}{2\lambda}\mathbb{E}_{q_t}[(\nabla^2\psi^{-1}\nabla\log\frac{q_t}{p})^\top P_{K_t,q_t}\nabla^2\psi^{-1}\nabla\log\frac{q_t}{p}],$$

*where* $\mathrm{MKSD}_{K_t}$ *is defined in Eq.* (15); $P_{K_t,q_t}: L^2(q_t) \to L^2(q_t)$ *is the kernel integral operator:* $(P_{K_t,q_t}\varphi)(\cdot) \triangleq \mathbb{E}_{q_t(\theta)}[K_t(\cdot,\theta)\varphi(\theta)]$ *for a general kernel* $K_t$ *and vector-valued function* $\varphi$ *on* $\Theta$, *and the stated equality holds whenever integration-by-parts is applicable.*

**Proposition 11.** *Suppose* $(\theta_t)_{t\geq 0}$ *follows the mirrored dynamics* (8) *with* $g_t$ *chosen to be* $g_{q_t,K_t}^*$ *as in* (10). *Then, the dissipation of* $\mathrm{KL}(q_t\|p)$ *is*

$$\frac{d}{dt}\mathrm{KL}(q_t\|p) = -\mathrm{MKSD}_{K_t}(q_t, p)^2.$$

**Proof**    The proof directly follows from Thm. 3 since the optimization problem there matches the definition of MKSD in (15). □

Therefore, when the Mirror Stein Log-Sobolev inequality holds, we have

$$\frac{\mathrm{d}}{\mathrm{d}t}\mathrm{KL}(q_t\|p) \leq -2\lambda\mathrm{KL}(q_t\|p),$$

and the exponential convergence $\mathrm{KL}(q_t\|p) \leq \mathrm{KL}(q_0\|p)e^{-2\lambda t}$ follows by Gronwall's lemma (Gronwall, 1919).

**Definition 3** (Mirror Stein Poincaré inequality). *We say that the distribution* $p$ *satisfies the Mirror Stein Poincaré inequality (cf., Duncan et al. 2019, Eq. (57)) with strongly convex* $\psi$ *and constant* $\lambda$ *if*

$$\mathrm{Var}_p[\phi] \leq \frac{1}{\lambda}\mathbb{E}_p[\nabla\phi^\top P_{K_t,p}\nabla^2\psi^{-1}\nabla\phi]$$

*for all* $\phi \in L^2(p) \cap C^\infty(\Theta)$ *that is locally Lipschitz, where* $P_{K_t,p}$ *is the kernel integral operator under* $p$ *defined similarly as in Definition 2.*

This inequality can also be viewed as a kernelized generalization of the mirror Poincaré inequality introduced in Chewi et al. (2020, Def. 1) for proving exponential convergence of mirror-Langevin diffusion. In a manner analogous to Thm. 1 of Chewi et al. (2020), the following proposition relates the Mirror Stein Poincaré inequality to chi-squared divergence.

**Proposition 12.** *Suppose* $(\theta_t)_{t\geq 0}$ *follows the mirrored dynamics* (8) *with* $g_t$ *chosen to be* $g_{q_t,K_t}^*$ *as in* (10). *Then, the dissipation of chi-square divergence* $\chi^2(q_t\|p)$ *is*

$$\frac{\mathrm{d}}{\mathrm{d}t}\chi^2(q_t\|p) = -2\mathbb{E}_{q_t}\left[\nabla\frac{q_t}{p}^\top P_{K_t,p}\nabla^2\psi^{-1}\nabla\frac{q_t}{p}\right]$$

*whenever integration-by-parts is applicable.*

**Proof**    We first note that by applying integration-by-parts, $g_{q_t,K_t}^*$ as in (10) can be equivalently written as

$$g_{q_t,K_t}^* = -P_{K_t,q_t}\nabla^2\psi^{-1}\nabla\log\frac{q_t}{p}.$$

Then using the Fokker-Planck equation of $q_t$ under the dynamics, we have

$$\frac{d}{dt}\chi^2(q_t\|p) = \frac{d}{dt}\int\left(\frac{q_t}{p}\right)^2 dp = 2\int\frac{q_t}{p}\cdot\frac{dq_t}{dt} = -2\mathbb{E}_{q_t}\left[g_{t,K_t}^{*\top}\nabla\frac{q_t}{p}\right]$$

$$= -2\mathbb{E}_{q_t}\left[\left(P_{K_t,q_t}\nabla^2\psi^{-1}\nabla\log\frac{q_t}{p}\right)^\top\nabla\frac{q_t}{p}\right] = -2\mathbb{E}_{q_t}\left[\nabla\frac{q_t}{p}^\top P_{K_t,p}\nabla^2\psi^{-1}\nabla\frac{q_t}{p}\right].$$

□

Note that the right hand side of the equation differs from the Mirror Stein Poincaré inequality only in the base measure of the expectation. Duncan et al. (2019) proposes to replace $q_t$ with $p$ to study the convergence near equilibrium (See their Sec. 6, where Eq. (46) is replaced with Eq. (51)). If we do the same and combine this identity with the Mirror Stein Poincaré inequality, we obtain $\frac{\mathrm{d}}{\mathrm{d}t}\chi^2(q_t\|p) \leq -2\lambda\mathrm{Var}_p[\frac{q_t}{p}] = -2\lambda\chi^2(q_t\|p)$, which implies exponential convergence in KL $\mathrm{KL}(q_t\|p) \leq \chi^2(q_t\|p) \leq \chi^2(q_0\|p)e^{-2\lambda t}$ by Gronwall's lemma (Gronwall, 1919).

# I PROOFS

## I.1 PROOF OF PROP. 1

**Proof** Fix any $g \in \mathcal{G}_\psi$. Since $g$ and $\nabla g$ are bounded and $\nabla^2 \psi(\theta)^{-1} \nabla \log p(\theta)$ and $\nabla \cdot \nabla^2 \psi(\theta)^{-1}$ are $p$-integrable, the expectation $\mathbb{E}_{\theta \sim p}[(\mathcal{M}_{p,\psi} g)(\theta)]$ exists. Because $\Theta$ is convex, $\Theta_r$ is bounded and convex with Lipschitz boundary. Since $p \nabla^2 \psi^{-1} g \in C^1$, we have

$$
\begin{aligned}
|\mathbb{E}_p[(\mathcal{M}_{p,\psi} g)(\theta)]| &= |\mathbb{E}_p[g(\theta)^\top \nabla^2 \psi(\theta)^{-1} \nabla \log p(\theta) + \nabla \cdot (\nabla^2 \psi(\theta)^{-1} g(\theta))]| \\
&= \left| \int_\Theta \nabla p(\theta)^\top \nabla^2 \psi(\theta)^{-1} g(\theta) + p(\theta) \nabla \cdot (\nabla^2 \psi(\theta)^{-1} g(\theta)) d\theta \right| \\
&= \left| \int_\Theta \nabla \cdot (p(\theta) \nabla^2 \psi(\theta)^{-1} g(\theta)) d\theta \right| \\
&= \left| \lim_{r \to \infty} \int_{\Theta_r} \nabla \cdot (p(\theta) \nabla^2 \psi(\theta)^{-1} g(\theta)) d\theta \right| \quad \text{(by dominated convergence)} \\
&= \left| \lim_{r \to \infty} \int_{\partial\Theta_r} (p(\theta) \nabla^2 \psi(\theta)^{-1} g(\theta))^\top n_r(\theta) d\theta \right| \quad \text{(by the divergence theorem)} \\
&\leq \lim_{r \to \infty} \int_{\partial\Theta_r} p(\theta) \|g(\theta)\|_2 \|\nabla^2 \psi(\theta)^{-1} n_r(\theta)\|_2 d\theta \quad \text{(by Cauchy-Schwarz)} \\
&\leq \|g\|_\infty \lim_{r \to \infty} \int_{\partial\Theta_r} p(\theta) \|\nabla^2 \psi(\theta)^{-1} n_r(\theta)\|_2 d\theta = 0 \quad \text{(by assumption).}
\end{aligned}
$$

$\square$

## I.2 PROOF OF THM. 3: OPTIMAL MIRROR UPDATES IN RKHS

**Proof** Let $e_i$ denote the standard basis vector of $\mathbb{R}^d$ with the $i$-th element being 1 and others being zeros. Since $m \in \mathcal{H}_K$, we have

$$
m(\theta)^\top \nabla^2 \psi(\theta)^{-1} \nabla \log p(\theta) = \langle m, K(\cdot, \theta) \nabla^2 \psi(\theta)^{-1} \nabla \log p(\theta) \rangle_{\mathcal{H}_K}
$$

$$
\begin{aligned}
\nabla \cdot (\nabla^2 \psi(\theta)^{-1} m(\theta)) &= \sum_{i=1}^d \nabla_{\theta_i}(m(\theta)^\top \nabla^2 \psi(\theta)^{-1} e_i) \\
&= \sum_{i=1}^d \langle m, \nabla_{\theta_i}(K(\cdot, \theta) \nabla^2 \psi(\theta)^{-1} e_i) \rangle_{\mathcal{H}_K} \\
&= \langle m, \nabla_\theta \cdot (K(\cdot, \theta) \nabla^2 \psi(\theta)^{-1}) \rangle_{\mathcal{H}_K},
\end{aligned}
$$

where we define the divergence of a matrix as a vector whose elements are the divergences of each row of the matrix. Then, we write (9) as

$$
\begin{aligned}
&- \mathbb{E}_{q_t}[m(\theta)^\top \nabla^2 \psi(\theta)^{-1} \nabla \log p(\theta) + \nabla \cdot (\nabla^2 \psi(\theta)^{-1} m(\theta))] \\
&= -\mathbb{E}_{q_t}[\langle m, K(\cdot, \theta) \nabla^2 \psi(\theta)^{-1} \nabla \log p(\theta) + \nabla_\theta \cdot (K(\cdot, \theta) \nabla^2 \psi(\theta)^{-1}) \rangle_{\mathcal{H}_K}] \\
&= -\langle m, \mathbb{E}_{q_t}[K(\cdot, \theta) \nabla^2 \psi(\theta)^{-1} \nabla \log p(\theta) + \nabla_\theta \cdot (K(\cdot, \theta) \nabla^2 \psi(\theta)^{-1})] \rangle_{\mathcal{H}_K} \\
&= -\langle m, \mathbb{E}_{q_t}[\mathcal{M}_{p,\psi} K(\cdot, \theta)] \rangle_{\mathcal{H}_K}.
\end{aligned}
$$

Therefore, the optimal direction in the $\mathcal{H}_K$ norm ball $\mathcal{B}_{\mathcal{H}_K} = \{g : \|g\|_{\mathcal{H}_K} \leq 1\}$ that minimizes (9) is $g_t^* \propto g_{q_t,K}^* = \mathbb{E}_{q_t}[\mathcal{M}_{p,\psi} K(\cdot, \theta)]$. $\square$

## I.3 PROOF OF THM. 4: MIRRORED SVGD UPDATES

**Proof** A p.d. kernel $k$ composed with any map $\phi$ is still a p.d. kernel. To prove this, let $\{x_1, \ldots, x_p\} = \{\phi(\eta_1), \ldots, \phi(\eta_n)\}$, $p \leq n$. Then

$$
\sum_{i,j} \alpha_i \alpha_j k(\phi(\eta_i), \phi(\eta_j)) = \sum_{\ell,m} \beta_\ell \beta_m k(x_\ell, x_m) \geq 0,
$$

where $\beta_\ell = \sum_{i \in S_\ell} \alpha_i$, $S_\ell = \{i : \phi(\eta_i) = x_\ell\}$. Therefore, $k_\psi(\eta, \eta') = k(\nabla\psi^*(\eta), \nabla\psi^*(\eta'))$ is a p.d. kernel. Plugging $K = kI$ into Lem. 13, for any $\theta' \in \Theta$ and $\eta' = \nabla\psi(\theta')$, we have

$$g^*_{q_t, K_k}(\theta') = \mathbb{E}_{\eta_t \sim q_{t,H}}[K_{\nabla\psi^*}(\nabla\psi(\theta'), \eta_t)\nabla \log p_H(\eta_t) + \nabla_{\eta_t} \cdot K_{\nabla\psi^*}(\nabla\psi(\theta'), \eta_t)]$$

$$= \mathbb{E}_{\eta_t \sim q_{t,H}}[k(\nabla\psi^*(\eta'), \nabla\psi^*(\eta_t))\nabla \log p_H(\eta_t) + \sum_{j=1}^d \nabla_{\eta_{t,j}} k(\nabla\psi^*(\eta'), \nabla\psi^*(\eta_t))e_j]$$

$$= \mathbb{E}_{\eta_t \sim q_{t,H}}[k_\psi(\eta', \eta_t)\nabla \log p_H(\eta_t) + \nabla_{\eta_t} k_\psi(\eta', \eta_t)].$$

$\square$

### I.4 PROOF OF PROP. 5: SINGLE-PARTICLE SVMD IS MIRROR DESCENT

**Proof** When $n = 1$, $\lambda_1 = k(\theta_t, \theta_t)$, $u_1 = 1$, and thus $K_{\psi,t}(\theta_t, \theta_t) = k(\theta_t, \theta_t)\nabla^2\psi(\theta_t)$. $\square$

### I.5 PROOF OF THM. 6: CONVERGENCE OF MIRRORED UPDATES AS $n \to \infty$

**Proof** The idea is to reinterpret our mirrored updates as one step of a matrix SVGD in $\eta$ space based on Lem. 13 and then follow the path of Gorham et al. (2020, Thm. 7). Assume that $q^n_{t,H}$ and $q^\infty_{t,H}$ have integrable means. Let $\eta^n, \eta^\infty$ be an optimal Wasserstein-1 coupling of $q^n_{t,H}$ and $q^\infty_{t,H}$. Let $\Phi_{q_t, K_t}$ denote the transform through one step of mirrored update: $\theta_t = \nabla\psi^\star(\eta_t)$, $\eta_{t+1} = \eta_t + \epsilon_t g^*_{q_t, K_t}(\theta_t)$. Then, with Lem. 13, we have

$$\|\Phi_{q_t, K_t}(\eta) - \Phi_{q_t, K_t}(\eta')\|_2$$
$$= \|\eta + \epsilon_t g^*_{q^n_t, K_t}(\theta) - \eta' - \epsilon_t g^*_{q^\infty_t, K_t}(\theta')\|_2$$
$$\leq \|\eta - \eta'\|_2 + \epsilon_t \|g^*_{q^n_t, K_t}(\theta) - g^*_{q^\infty_t, K_t}(\theta')\|_2$$
$$\leq \|\eta - \eta'\|_2$$
$$+ \epsilon_t \|\mathbb{E}_{\eta^n}[K_{\nabla\psi^*, t}(\eta, \eta^n)\nabla \log p_H(\eta^n) + \nabla_{\eta^n} \cdot K_{\nabla\psi^*, t}(\eta, \eta^n)$$
$$\quad - (K_{\nabla\psi^*, t}(\eta', \eta^n)\nabla \log p_H(\eta^n) + \nabla_{\eta^n} \cdot K_{\nabla\psi^*, t}(\eta', \eta^n))]\|_2$$
$$+ \epsilon_t \|\mathbb{E}_{\eta^n, \eta^\infty}[K_{\nabla\psi^*, t}(\eta', \eta^n)\nabla \log p_H(\eta^n) + \nabla_{\eta^n} \cdot K_{\nabla\psi^*, t}(\eta, \eta^n)$$
$$\quad - (K_{\nabla\psi^*, t}(\eta', \eta^\infty)\nabla \log p_H(\eta^\infty) + \nabla_{\eta^\infty} \cdot K_{\nabla\psi^*, t}(\eta', \eta^\infty))]\|_2$$
$$\leq \|\eta - \eta'\|_2 + \epsilon_t c_1(1 + \mathbb{E}[\|\eta^n\|_2])\|\eta - \eta'\|_2 + \epsilon_t c_2(1 + \|\eta'\|_2)\mathbb{E}_{\eta^n, \eta^\infty}[\|\eta^n - \eta^\infty\|_2]$$
$$= \|\eta - \eta'\|_2 + \epsilon_t c_1(1 + \mathbb{E}_{q^n_{t,H}}[\|\cdot\|_2])\|\eta - \eta'\|_2 + \epsilon_t c_2(1 + \|\eta'\|_2)W_1(q^n_{t,H}, q^\infty_{t,H}).$$

Since $\Phi_{q_t, K_t}(\eta^n) \sim q^n_{t+1,H}$, $\Phi_{q_t, K}(\eta^\infty) \sim q^\infty_{t+1,H}$, we conclude

$$W_1(q^n_{t+1,H}, q^\infty_{t+1,H})$$
$$\leq \mathbb{E}[\|\Phi_{q_t, K}(\eta^n) - \Phi_{q_t, K}(\eta^\infty)\|_2]$$
$$\leq (1 + \epsilon_t c_1(1 + \mathbb{E}_{q^n_{t,H}}[\|\cdot\|_2]))\mathbb{E}[\|\eta^n - \eta^\infty\|_2] + \epsilon_t c_2(1 + \|\eta'\|_2)W_1(q^n_{t,H}, q^\infty_{t,H})]$$
$$\leq (1 + \epsilon_t c_1(1 + \mathbb{E}_{q^n_{t,H}}[\|\cdot\|_2]) + \epsilon_t c_2(1 + \mathbb{E}_{q^\infty_{t,H}}[\|\cdot\|_2]))W_1(q^n_{t,H}, q^\infty_{t,H}).$$

The final claim $q^n_t \Rightarrow q^\infty_t$ now follows by the continuous mapping theorem as $\nabla\psi^*$ is continuous. $\square$

### I.6 PROOF OF THM. 7: INFINITE-PARTICLE MIRRORED STEIN UPDATES DECREASE KL AND MKSD

**Proof** Let $T_{q^\infty_t, K_t}$ denote transform of the density function through one step of mirrored update: $\theta_t = \nabla\psi^\star(\eta_t)$, $\eta_{t+1} = \eta_t + \epsilon_t g^*_{q^\infty_t, K_t}(\theta_t)$. Then

$$\text{KL}(q^\infty_{t+1}\|p) - \text{KL}(q^\infty_t\|p)$$
$$= \text{KL}(q^\infty_t\|T^{-1}_{q^\infty_t, K_t}p) - \text{KL}(q^\infty_t\|p)$$
$$= \mathbb{E}_{\eta_t \sim q^\infty_{t,H}}[\log p_H(\eta_t) - \log p_H(\eta_t + \epsilon_t g^*_{q^\infty_t, K_t}(\theta_t)) - \log|\det(I + \epsilon_t \nabla_{\eta_t} g^*_{q^\infty_t, K_t}(\theta_t))|],$$

where we have used the invariance of KL divergence under reparameterization: $\mathrm{KL}(q_t\|p) = \mathrm{KL}(q_{t,H}\|p_H)$. Following Liu (2017), we bound the difference of the first two terms as

$$\log p_H(\eta_t) - \log p_H(\eta_t + \epsilon_t g^*_{q_t^\infty, K_t}(\theta_t))$$

$$= -\int_0^1 \nabla_s \log p_H(\eta_t(s)) \, ds, \quad \text{where } \eta_t(s) \triangleq \eta_t + s\epsilon_t g^*_{q_t^\infty, K_t}(\theta_t)$$

$$= -\int_0^1 \nabla \log p_H(\eta_t(s))^\top (\epsilon_t g^*_{q_t^\infty, K_t}(\theta_t)) \, ds$$

$$= -\epsilon_t \nabla \log p_H(\eta_t)^\top g^*_{q_t^\infty, K_t}(\theta_t) + \int_0^1 (\nabla \log p_H(\eta_t) - \nabla \log p_H(\eta_t(s)))^\top (\epsilon_t g^*_{q_t^\infty, K_t}(\theta_t)) \, ds$$

$$\leq -\epsilon_t \nabla \log p_H(\eta_t)^\top g^*_{q_t^\infty, K_t}(\theta_t) + \epsilon_t \int_0^1 \|\nabla \log p_H(\eta) - \nabla \log p_H(\eta_t(s))\|_2 \cdot \|g^*_{q_t^\infty, K_t}(\theta_t)\|_2 \, ds$$

$$\leq -\epsilon_t \nabla \log p_H(\eta_t)^\top g^*_{q_t^\infty, K_t}(\theta_t) + \frac{L\epsilon_t^2}{2} \|g^*_{q_t^\infty, K_t}(\theta_t)\|_2^2,$$

and bound the log determinant term using Lem. 15:

$$-\log|\det(I + \epsilon_t \nabla_{\eta_t} g^*_{q_t^\infty, K_t}(\theta_t))| \leq -\epsilon_t \operatorname{Tr}(\nabla_{\eta_t} g^*_{q_t^\infty, K_t}(\theta_t)) + 2\epsilon_t^2 \|\nabla_{\eta_t} g^*_{q_t^\infty, K_t}(\theta_t)\|_F^2.$$

The next thing to notice is that $\mathbb{E}_{\eta_t \sim q_{t,H}^\infty}[\nabla \log p_H(\eta_t)^\top g^*_{q_t^\infty, K_t}(\theta_t) + \operatorname{Tr}(\nabla_{\eta_t} g^*_{q_t^\infty, K_t}(\theta_t))]$ is the square of the MKSD in (15). We can show this equivalence using the identity proved in Lem. 14:

$$\mathbb{E}_{\eta_t \sim q_{t,H}^\infty}[g^*_{q_t^\infty, K_t}(\theta_t)^\top \nabla \log p_H(\eta_t) + \operatorname{Tr}(\nabla_{\eta_t} g^*_{q_t^\infty, K_t}(\theta_t))]$$

$$= \mathbb{E}_{\theta_t \sim q_t^\infty}[g^*_{q_t^\infty, K_t}(\theta_t)^\top \nabla^2 \psi(\theta_t)^{-1} \nabla_{\theta_t}(\log p(\theta_t) - \log \det \nabla^2 \psi(\theta_t))$$

$$\quad + \operatorname{Tr}(\nabla^2 \psi(\theta_t)^{-1} \nabla g^*_{q_t^\infty, K_t}(\theta_t))]$$

$$= \mathbb{E}_{\theta_t \sim q_t^\infty}[g^*_{q_t^\infty, K_t}(\theta_t)^\top \nabla^2 \psi(\theta_t)^{-1} \nabla \log p(\theta_t) + \nabla \cdot (\nabla^2 \psi(\theta_t)^{-1} g^*_{q_t^\infty, K_t}(\theta_t))] \quad \text{(Lem. 14)}$$

$$= \mathbb{E}_{\theta_t \sim q_t^\infty}[(\mathcal{M}_{p,\psi} g^*_{q_t^\infty, K_t})(\theta_t)]$$

$$= \mathrm{MKSD}_{K_t}(q_t^\infty, p)^2.$$

Finally, we are going to bound $\|g^*_{q_t^\infty, K_t}(\theta_t)\|_2^2$ and $\|\nabla_{\eta_t} g^*_{q_t^\infty, K_t}(\theta_t)\|_F^2$. From the assumptions we have $\psi$ is $\alpha$-strongly convex and thus $\psi^*$ is $\frac{1}{\alpha}$-strongly smooth (Kakade et al., 2009), therefore $\|\nabla^2 \psi^*(\cdot)\|_2 \leq \frac{1}{\alpha}$. By Lem. 16, we know

$$\|g^*_{q_t^\infty, K_t}(\theta_t)\|_2^2 \leq \|g^*_{q_t^\infty, K_t}\|_{\mathcal{H}_{K_t}}^2 \|K(\theta_t, \theta_t)\|_{\mathrm{op}} = \mathrm{MKSD}_{K_t}(q_t^\infty, p)^2 \|K_t(\theta_t, \theta_t)\|_{\mathrm{op}},$$

$$\|\nabla_{\eta_t} g^*_{q_t^\infty, K_t}(\theta_t)\|_F^2 = \|\nabla^2 \psi^*(\eta_t) \nabla g^*_{q_t^\infty, K_t}(\theta_t)\|_F^2$$

$$\leq \|\nabla^2 \psi^*(\eta_t)\|_2^2 \|\nabla g^*_{q_t^\infty, K_t}(\theta_t)\|_F^2$$

$$\leq \frac{1}{\alpha^2} \|g^*_{q_t^\infty, K_t}\|_{\mathcal{H}_{K_t}}^2 \sum_{i=1}^d \|\nabla^2_{i, d+i} K_t(\theta_t, \theta_t)\|_{\mathrm{op}}$$

$$= \frac{1}{\alpha^2} \mathrm{MKSD}_{K_t}(q_t^\infty, p)^2 \sum_{i=1}^d \|\nabla^2_{i, d+i} K_t(\theta_t, \theta_t)\|_{\mathrm{op}},$$

where $\nabla^2_{i, d+i} K(\theta, \theta)$ denotes $\nabla^2_{\theta_i, \theta'_i} K(\theta, \theta')|_{\theta'=\theta}$. Combining all of the above, we have

$$\mathrm{KL}(q_{t+1}^\infty \| p) - \mathrm{KL}(q_t^\infty \| p)$$

$$\leq -\left( \epsilon_t - \frac{L\epsilon_t^2}{2} \sup_\theta \|K_t(\theta, \theta)\|_{\mathrm{op}} - \frac{2\epsilon_t^2}{\alpha^2} \sum_{i=1}^d \sup_\theta \|\nabla^2_{i, d+i} K_t(\theta, \theta)\|_{\mathrm{op}} \right) \mathrm{MKSD}_{K_t}(q_t^\infty, p)^2.$$

Plugging in the definition of $\kappa_1$ and $\kappa_2$ finishes the proof. $\qquad\square$

## I.7 PROOF OF THM. 8: $\text{MKSD}_{K_k}$ DETERMINES WEAK CONVERGENCE

**Proof** According to Thm. 4,

$$g_{q,K_k}^* = \mathbb{E}_{q_H}[k(\cdot, \nabla\psi^*(\eta))\nabla \log p_H(\eta) + \nabla_\eta k(\nabla\psi^*(\eta), \cdot)],$$

where $q_H(\eta)$ denotes the density of $\eta = \nabla\psi(\theta)$ under the distribution $\theta \sim q$. From the assumptions we have $k(\theta, \theta') = \kappa(\nabla\psi(\theta), \nabla\psi(\theta'))$. With this specific choice of $k$, the squared MKSD is

$$
\begin{aligned}
\text{MKSD}_{K_k}(q, p)^2 &= \|g_{q,K_k}^*\|_{\mathcal{H}_{K_k}}^2 \\
&= \mathbb{E}_{\eta, \eta' \sim q_H}\left[\frac{1}{p_H(\eta)p_H(\eta')}\nabla_\eta\nabla_{\eta'}(p_H(\eta)k(\nabla\psi^*(\eta), \nabla\psi^*(\eta'))p_H(\eta'))\right] \\
&= \mathbb{E}_{\eta, \eta' \sim q_H}\left[\frac{1}{p_H(\eta)p_H(\eta')}\nabla_\eta\nabla_{\eta'}(p_H(\eta)\kappa(\eta, \eta')p_H(\eta'))\right].
\end{aligned}
\tag{21}
$$

The final expression in (21) is the squared kernel Stein discrepancy (KSD) (Liu et al., 2016; Chwialkowski et al., 2016; Gorham & Mackey, 2017) between $q_H$ and $p_H$ with the kernel $\kappa$: $\text{KSD}_\kappa(q_H, p_H)^2$. Recall that it is proved in Gorham & Mackey (2017, Theorem 8) that, for $\kappa(x, y) = (c^2 + \|x - y\|_2^2)^\beta$ with $\beta \in (-1, 0)$ and distantly dissipative $p_H$ with Lipschitz score functions, $q_H \Rightarrow p_H$ if $\text{KSD}_\kappa(q_H, p_H) \to 0$. The advertised result ($q \Rightarrow p$ if $\text{MKSD}_{K_k}(q, p) \to 0$) now follows by the continuous mapping theorem as $\nabla\psi^*$ is continuous. $\qquad\square$

## J LEMMAS

**Lemma 13.** *Let $K_{\nabla\psi^*}(\eta, \eta') \triangleq K(\nabla\psi^*(\eta), \nabla\psi^*(\eta'))$. The mirrored updates $g_{q_t,K}^*$ in (10) can be equivalently expressed as*

$$g_{q_t,K}^* = \mathbb{E}_{q_{t,H}}[K_{\nabla\psi^*}(\nabla\psi(\cdot), \eta)\nabla \log p_H(\eta) + \nabla_\eta \cdot K_{\nabla\psi^*}(\nabla\psi(\cdot), \eta)].$$

**Proof** We will use the identity proved in Lem. 14.

$$
\begin{aligned}
g_{q_t,K}^* &= \mathbb{E}_{q_t}[\mathcal{M}_{p,\psi}K(\cdot, \theta)] \\
&= \mathbb{E}_{q_t}[K(\cdot, \theta)\nabla^2\psi(\theta)^{-1}\nabla \log p(\theta) + \nabla_\theta \cdot (K(\cdot, \theta)\nabla^2\psi(\theta)^{-1})] \\
&= \mathbb{E}_{q_t}[K(\cdot, \theta)\nabla^2\psi(\theta)^{-1}\nabla_\theta(\log p_H(\nabla\psi(\theta)) + \log\det\nabla^2\psi(\theta)) + \nabla_\theta \cdot (K(\cdot, \theta)\nabla^2\psi(\theta)^{-1})] \\
&\qquad\qquad\qquad\qquad\qquad\qquad\qquad\qquad\qquad\qquad\qquad\text{(by change-of-variable formula)} \\
&= \mathbb{E}_{q_t}\Big[K(\cdot, \theta)\nabla^2\psi(\theta)^{-1}\nabla_\theta \log p_H(\nabla\psi(\theta)) + \sum_{i,j=1}^d [\nabla^2\psi(\theta)^{-1}]_{ij}\nabla_{\theta_i}K(\cdot, \theta)_{:,j}\Big] \\
&\qquad\qquad\qquad\qquad\qquad\qquad\qquad\qquad\qquad\text{(by applying Lem. 14 to each row of $K(\cdot, \theta)$)} \\
&= \mathbb{E}_{q_t}\Big[K(\cdot, \theta)\nabla^2\psi(\theta)^{-1}\nabla_\theta \log p_H(\nabla\psi(\theta)) + \sum_{j=1}^d \nabla_{\eta_j}K(\cdot, \theta)_{:,j}\Big] \\
&= \mathbb{E}_{q_{t,H}}\Big[K(\cdot, \nabla\psi^*(\eta))\nabla \log p_H(\eta) + \sum_{j=1}^d \nabla_{\eta_j}K(\cdot, \nabla\psi^*(\eta))_{:,j}\Big] \\
&= \mathbb{E}_{q_{t,H}}[K_{\nabla\psi^*}(\nabla\psi(\cdot), \eta)\nabla \log p_H(\eta) + \nabla_\eta \cdot K_{\nabla\psi^*}(\nabla\psi(\cdot), \eta)],
\end{aligned}
$$

where $A_{:,j}$ denotes the $j$-th column of a matrix $A$. $\qquad\square$

**Lemma 14.** *For a strictly convex function $\psi \in C^2 : \mathbb{R}^d \to \mathbb{R}$ and any vector-valued $g \in C^1 : \mathbb{R}^d \to \mathbb{R}^d$, the following relation holds:*

$$\nabla \cdot (\nabla^2\psi(\theta)^{-1}g(\theta)) = \text{Tr}(\nabla^2\psi(\theta)^{-1}\nabla g(\theta)) - g(\theta)^\top\nabla^2\psi(\theta)^{-1}\nabla_\theta \log\det\nabla^2\psi(\theta).$$

**Proof** By the product rule of differentiation:

$$\nabla \cdot (\nabla^2\psi(\theta)^{-1}g(\theta)) = \text{Tr}(\nabla^2\psi(\theta)^{-1}\nabla g(\theta)) + g(\theta)^\top\nabla \cdot (\nabla^2\psi(\theta)^{-1}). \tag{22}$$

This already gives us the first term on the right side. Next, we have

$$[\nabla^2 \psi(\theta)^{-1} \nabla \log \det \nabla^2 \psi(\theta)]_i$$

$$= \sum_{j=1}^{d} [\nabla^2 \psi(\theta)^{-1}]_{ij} \operatorname{Tr}(\nabla^2 \psi(\theta)^{-1} \nabla_{\theta_j} \nabla^2 \psi(\theta))$$

$$= \sum_{j=1}^{d} [\nabla^2 \psi(\theta)^{-1}]_{ij} \sum_{\ell,m=1}^{d} [\nabla^2 \psi(\theta)^{-1}]_{\ell m} [\nabla_{\theta_j} \nabla^2 \psi(\theta)]_{m\ell}$$

$$= \sum_{j,\ell,m=1}^{d} [\nabla^2 \psi(\theta)^{-1}]_{ij} [\nabla^2 \psi(\theta)^{-1}]_{\ell m} \nabla_{\theta_j} \nabla^2 \psi(\theta)_{m\ell}$$

$$= \sum_{j,\ell,m=1}^{d} [\nabla^2 \psi(\theta)^{-1}]_{ij} \nabla_{\theta_m} \nabla^2 \psi(\theta)_{j\ell} [\nabla^2 \psi(\theta)^{-1}]_{\ell m}$$

$$= - \sum_{m=1}^{d} \nabla_{\theta_m} (\nabla^2 \psi(\theta)^{-1})_{im}$$

$$= -[\nabla \cdot \nabla^2 \psi(\theta)^{-1}]_i.$$

Plugging the above relation into (22) proves the claimed result. $\qquad\square$

**Lemma 15** (Liu, 2017, Lemma A.1). *Let $A$ be a square matrix, and $0 < \epsilon < \frac{1}{2\|A + A^\top\|_{\mathrm{op}}}$. Then,*

$$\log |\det(I + \epsilon A)| \geq \epsilon \operatorname{Tr}(A) - 2\epsilon^2 \|A\|_F^2,$$

*where $\|\cdot\|_F$ denotes the Frobenius norm of a matrix.*

**Lemma 16.** *Let $K$ be a matrix-valued kernel and $\mathcal{H}_K$ be the corresponding RKHS. Then, for any $f \in \mathcal{H}_K$ ($f$ is vector-valued), we have*

$$\|f(x)\|_2 \leq \|f\|_{\mathcal{H}_K} \|K(x,x)\|_{\mathrm{op}}^{1/2}, \quad \|\nabla f(x)\|_F^2 \leq \|f\|_{\mathcal{H}_K}^2 \sum_{i=1}^{d} \|\nabla_{x_i,x_i'}^2 K(x,x')|_{x'=x}\|_{\mathrm{op}},$$

*where $\|\cdot\|_{\mathrm{op}}$ denotes the operator norm of a matrix induced by the vector 2-norm.*

**Proof** We first bound the $\|f(x)\|_2$ as

$$\|f(x)\|_2 = \sup_{\|y\|_2=1} f(x)^\top y = \sup_{\|y\|_2=1} \langle f, K(\cdot,x)y \rangle_{\mathcal{H}_K} \leq \|f\|_{\mathcal{H}_K} \sup_{\|y\|_2=1} \|K(\cdot,x)y\|_{\mathcal{H}_K}$$

$$= \|f\|_{\mathcal{H}_K} \sup_{\|y\|_2=1} (y^\top K(x,x)y)^{1/2} \leq \|f\|_{\mathcal{H}_K} \sup_{\|y\|_2=1} \sup_{\|u\|_2=1} (u^\top K(x,x)y)^{1/2}$$

$$= \|f\|_{\mathcal{H}_K} \sup_{\|y\|_2=1} \|K(x,x)y\|_2^{1/2} = \|f\|_{\mathcal{H}_K} \|K(x,x)\|_{\mathrm{op}}^{1/2}.$$

The second result follows similarly,

$$\|\nabla f(x)\|_F^2 = \sum_{i=1}^{d} \|\nabla_{x_i} f(x)\|_2^2 = \sum_{i=1}^{d} \sup_{\|y\|_2=1} (\nabla_{x_i} f(x)^\top y)^2 = \sum_{i=1}^{d} \sup_{\|y\|_2=1} (\langle f, \nabla_{x_i} K(\cdot,x)y \rangle_{\mathcal{H}_K})^2$$

$$\leq \|f\|_{\mathcal{H}_K}^2 \sum_{i=1}^{d} \sup_{\|y\|_2=1} \|\nabla_{x_i} K(\cdot,x)y\|_{\mathcal{H}_K}^2 = \|f\|_{\mathcal{H}_K}^2 \sum_{i=1}^{d} \sup_{\|y\|_2=1} (y^\top \nabla_{x_i,x_i'}^2 K(x,x')|_{x=x'} y)$$

$$\leq \|f\|_{\mathcal{H}_K}^2 \sum_{i=1}^{d} \sup_{\|y\|_2=1} \sup_{\|u\|_2=1} (u^\top \nabla_{x_i,x_i'}^2 K(x,x')|_{x=x'} y)$$

$$= \|f\|_{\mathcal{H}_K}^2 \sum_{i=1}^{d} \|\nabla_{x_i,x_i'}^2 K(x,x')|_{x'=x}\|_{\mathrm{op}}.$$

$\qquad\square$

