# OpenReview forum: "Sampling with Mirrored Stein Operators"
_ICLR.cc/2022/Conference — ICLR 2022 Spotlight_

### Official Review · Reviewer_5TzE · 2021-10-21

**Correctness:** 4
**Technical Novelty And Significance:** 3
**Empirical Novelty And Significance:** 3
**Recommendation:** 8
**Confidence:** 3

**Main Review:**

Strengths:
- The paper is dense and offers a large number of both algorithmic and theoretical contributions. The use of mirror transformations to build specialized Stein's operators show a remarkable technical mastery.
- The addressed problems are relevant. While several prior works have already addressed the Riemannian case, the convergence properties of this method seems to be superior. The application to constrained problems is novel and in my opinion particularly useful.
- The writing is clear. Both prior work and contributions are clearly explained.
- Good balance between theoretical and algorithmic contributions.

Weaknesses:
- The abundance of content makes the overall narrative difficult to follow. the paper introduces several related methods in order to solve several loosely related problems. I would have appreciated a more focused presentation.
- The experiments are unfocused, with too little space devoted to each application. I would have preferred to see less experiments with more detailed discussion and analysis.

**Summary Of The Paper:**

the paper introduces a new family of deterministic particle samplers which apply mirror transformations to the Stein's variational gradient descent method. This approach can be used to generalize SVGD to both constrained and non-euclidean settings. The authors introduce two algorithms of this type that differ by the choice of the kernel function. One of these, SVMD is shown to reduce to regular mirror descent when a single particle is used.

**Summary Of The Review:**

The contribution is novel and solid with a wide range of potential applications. The paper is slightly too dense and somewhat unfocused. However, the main contributions are presented in a clear and understandable way.

---

> ### Author Response · Authors · 2021-11-22
> **Response**
>
> Thank you for the time you’ve taken to review our work and for the positive and constructive feedback!  We are glad that you found the addressed problems relevant, the solutions novel and useful, and the writing clear.  In an attempt to make the paper more accessible and focused, in the latest revision we have moved some inessential details from the main paper to the appendix.  In addition, due to the space limit, we have included additional results and more detailed analysis of the experiments in Appendix E.

---

> > ### Comment · Reviewer_5TzE · 2021-11-29
> > **Acknowledgment**
> >
> > Dear authors,
> >
> > I wish to thank you for the additional work on the manuscript. I am happy with the revision and I will therefore keep my original score.

---

### Official Review · Reviewer_Zj9c · 2021-11-01

**Correctness:** 4
**Technical Novelty And Significance:** 3
**Empirical Novelty And Significance:** 3
**Recommendation:** 10
**Confidence:** 3

**Main Review:**

The paper tackles the important problem of non-Euclidean or constrained space with particle methods. Its significance is therefore very high.
The paper is very complete and introduces all the different notions needed to understand the theory of the proposed methods.
The only comment I would have is that with the appendix being itself ~14 pages, this submission would better fit as a journal paper as it is.

Questions/Comments:
- How does the algorithm compare with vanilla SVGD combined with a bijection to solve the constraint problem (for example x->exp(x) for positive real values)?
- Could you explain why "the modes of the mirrored density $p_H(η)$ need not match those of the target density $p(θ)$"?
- Could you comment on the cost of computing eigenfunctions and eigenvalues for SVMD?

**Summary Of The Paper:**

The paper introduces new methods to run SVGD in constrained domains and non-euclidean geometries.
The authors develop the theory to combine mirror descent dynamics with the Stein's method via what they call mirrored Stein operators.
They also show experimental results on two concrete problems and prove convergence guarantees.

**Summary Of The Review:**

This paper addresses an interesting problem and its development looks very exciting.
The work is very complete and addresses both theoretical and experimental aspects of the proposed methods.
The theory proposed looks sound with a strong background and the experiments look very convincing.

---

> ### Author Response · Authors · 2021-11-22
> **Response**
>
> Thank you for the time you’ve taken to review our work and for the positive and constructive feedback!  We are glad that you found the problem tackled important, the significance very high, and the paper very complete.  We respond to each of your remaining comments below.
>
> ### Comparison with vanillaSVGD combined with a bijection
> The MSVGD algorithm is identical to running SVGD in the $\\eta$ space with a special composition kernel $k’(\\eta, \\eta’) = k(\\nabla\\psi^*(\\eta), \\nabla\\psi^*(\\eta’))$.  Specifically, the map $x \\mapsto \\exp(x)$ can be recovered by choosing $\\psi(\\theta) = \\sum_{j=1}^d \\theta_j \\log \\theta_j - \\theta_j$.
>
>
> ### Why the modes of the mirrored density of $\\eta$ need not match those of the target density $\\theta$?
> Appendix E in the latest revision includes two simple concrete examples in which the modes of the mirrored density do not match those of the target density.  Each example is a one-dimensional Beta distribution supported on [0, 1].  In one case, $p_H(\\eta)$ is unimodal, and its unique point of highest density maps to the point of **smallest** density for $p(\\theta)$.  In the other case, both $p(\\theta)$ and $p_H(\\eta)$ are unimodal, but the mode of  $p(\\theta)$ does not map to the mode of $p_H(\\eta)$.  From a theoretical standpoint, a mode of $p_H(\\eta)$ need not match that of $p(\\theta)$ due to the presence of the log determinant term in the change-of-variable formula:
> $
> \\log p_H(\\eta) = \\log p(\\nabla \\psi^*(\\eta)) + \\log \\det \\nabla^2 \\nabla\\psi^*(\\eta).
> $
>
>
> ### The cost of computing eigenfunctions and eigenvalues for SVMD
> As explained in section 4.3, due to the particle representation of $q_t$, we only need to solve a matrix eigenvalue problem, which costs $O(n^3)$ time and $O(n^2)$ memory. In practice the number of particles used for particle evolution algorithms is relatively small, even for SVGD, due to the $O(n^2)$ cost of updates. We have produced a practical SVMD implementation that is computationally competitive with MSVGD and SVGD for standard particle counts $n$. For example, in our post-selection inference experiment, both SVMD and MSVGD are very fast: SVMD takes ~10s total to generate 5000 points in batches of size $n=50$, while MSVGD takes ~3s.

---

### Official Review · Reviewer_BkpK · 2021-11-04

**Correctness:** 3
**Technical Novelty And Significance:** 4
**Empirical Novelty And Significance:** 4
**Recommendation:** 6
**Confidence:** 4

**Main Review:**

[Strengths]
The paper is overall well-written and makes reasonable attempts to provide sufficient background to understand the proposed algorithms. The literature review is thorough.

[Weakness]
- In the discussion in this work, constrained domains seem to be restricted to simplex. I wonder if MSVGD and SVMD are applicable to constrained domains in general, for example, a conjunction of non-overlapping simplexes and whether the convergence analysis still holds in this case.

- At the end of Section 4.2, I find the statement that MSVGD does not necessarily match the modes of the target density lacks theoretical or empirical evidence. In the multimodal sparse Dirichlet experiments, MSVGD and SVMD have pretty similar performances. Since this statement is the motivation for SVMD, it would be more convincing to show why or when SVMD is able to capture the modes while MSVGD is not either theoretically or empirically. Also, a more detailed comparison of these two algorithms would be appreciated.


[Minor Comments]
- Some references are not properly presented. There are some cases where \citep should have been used but \citet is used; the opposite cases also happen. A careful pass on citations might be beneficial.
- In Figure 2, the curves for MSVGD and SVMD are partially invisible when they are close to 0.0.

**Summary Of The Paper:**

The aim of this work is to extend the existing particle evolution method Stein variational gradient descent or SVGD to constrained domains and non-Euclidean geometry. Three algorithms are proposed for this purpose based on mirrored Stein operators. The first is Stein variational mirror descent that runs SVGD in dual space s.t. the updated particles stay in constrained domains. The second one is Stein variational mirror descent defined with some adaptive kernels and it is also applicable to constrained domain problems. The third one is Stein variational natural gradient which is intended for unconstrained problems with informative metric tensors. Empirical and convergence analyses are further provided for all three proposed algorithms.

**Summary Of The Review:**

The theoretical contribution of this work is solid. Still, quality and clarity need to be improved in several points as mentioned in main review.

---

> ### Author Response · Authors · 2021-11-17
> **A conjunction of non-overlapping simplexes**
>
> Dear Reviewer,
>
> Could you clarify what you mean by "a conjunction of non-overlapping simplexes"? We understand it as
> $
> \\{x \\in \\mathbb{R}^{K_1+K_2}| x_{1:K_1} \\in \\Delta^{K_1}, x_{K_1+1:K_1+K_2} \\in \\Delta^{K_2}\\}
> $,
> where $\\Delta^K$ denotes a K-simplex. Is that correct?
>
> Best,
> Authors

---

> > ### Comment · Reviewer_BkpK · 2021-11-17
> > **Clarification**
> >
> > Nope. By a conjunction of non-overlapping simplexes, I mean for example \{ $x \in \mathbb{R}^K \mid x \in \Delta_1 \cup \Delta_2, \Delta_1 \cap \Delta_2 = \emptyset $\}. The simplest example could be in one-dimensional case, a variable $x$ has its constrained domain to be \{$ x \in [0, 1] \cup [2, 3] $\}.

---

> ### Author Response · Authors · 2021-11-22
> **Response**
>
> Thank you for answering our question and for the positive and constructive feedback!  We are glad that you found our paper well-written, our contributions significant, and our literature review thorough.  We respond to each of your remaining comments below.
>
> ### Constrained domains
> As stated at the beginning of section 3 and like mirror descent, our algorithms require the domain $\\Theta$ to be convex and so do not directly apply to non-convex domains with disjoint components (like the union of two non-overlapping simplices). However, our approach is by no means limited to simplices.  For example, in the post selection inference task (section 5.2) the $\\Theta$ is not a simplex but rather the nonnegative orthant with $\\psi(\\theta) = \\sum_{j=1}^d (\\theta_j \\log \\theta_j - \\theta_j)$. Moreover, our algorithms are applicable to any convex domain with an appropriate mirror map, and such maps have been developed, for instance, for orthants in $\\mathbb{R}^d$, rectangles in $\\mathbb{R}^d$ (i.e., the product of 2-simplices), the set of positive definite matrices with $\\psi$ the matrix entropy, and spectraplices with $\\psi$ the Von Neumann entropy.
>
> ### When MSVGD fails to capture target modes
> Thank you for this suggestion!  Appendix E in the latest revision includes two simple concrete examples in which the modes found by single-particle MSVGD do not match those of the target density.  Each example is a one-dimensional Beta distribution supported on [0, 1].  In one case, $p_H(\\eta)$ is unimodal, and its unique point of highest density maps to the point of **smallest** density for $p(\\theta)$.  In the other case, both $p(\\theta)$ and $p_H(\\eta)$ are unimodal, but the mode of  $p(\\theta)$ does not map to the mode of $p_H(\\eta)$.  In either case, single-particle MSVGD will converge to an inappropriate value as it simply performs gradient ascent on $p_H(\\eta)$. From a theoretical standpoint, a mode of  $p_H(\\eta)$ need not match that of $p(\\theta)$ due to the presence of the log determinant term in the change-of-variable formula:
> $
> \\log p_H(\\eta) = \\log p(\\nabla \\psi^*(\\eta)) + \\log \\det \\nabla^2 \\psi^*(\\eta).
> $
>
> To verify the theoretical argument, we ran MSVGD and SVMD with n=1 on both examples. For the first example, MSVGD returns 0.499 (eta value -0.004, near zero --- the mode of $p_H(\\eta)$); SVMD returns 0.002 (near zero ---  one mode of $p(\\theta)$). For the second example, MSVGD returns 0.1 (eta value -2.2, near the mode of $p_H(\\eta)$); SVMD returns 0.011 (the mode of $p(\\theta)$).

---

> > ### Comment · Reviewer_BkpK · 2021-12-01
> > **Acknowledgement**
> >
> > I would like to thank the authors for the response which answers most of my questions.

---

> > > ### Author Response · Authors · 2021-12-01
> > > **Other concerns?**
> > >
> > > Dear Reviewer,
> > >
> > > Thanks for reading the rebuttal. We are glad that you find it useful. As indicated by the rating, do you have any more concerns about the draft that we could try to answer?
> > >
> > > Best,
> > > Authors

---

### Official Review · Reviewer_dVdw · 2021-11-07

**Correctness:** 3
**Technical Novelty And Significance:** 3
**Empirical Novelty And Significance:** 3
**Recommendation:** 8
**Confidence:** 3

**Main Review:**

Strengths:
1. The premise of the paper is clear and interesting : in traditional optimization, mirror descent allows you to optimize over constrained feasible sets, and this paper attempts a similar approach to sampling.

1. In order to achieve the sampling analogue of mirror descent, the authors define an infinitesimal generator for a Markov process that has an equilibrium density $ p $. Since $ p $ has constraints (for example, it should be a sparse Dirichlet distribution), the authors construct the generators via the mirror maps associated with the constraints. I think the ideas used to construct these generators are quite non-trivial.

1. The authors show that their algorithm mixes quicker in simulated experiments.

Weaknesses:
1. The writing can be improved. The authors try to give a lot of information about gradient flows that are often repetitive and not particularly friendly to readers. For example, the paragraph between Eqn (4) and (5) will not help an unfamiliar reader. Additionally, I did not understand the details of the experiments in section 5.2

1. Some of the notation is quite confusing -- the authors use $ \theta $ and $\theta_t$ interchangeably at times. This is not a problem in some cases, like Eqn (10), but Eqn (11) is very confusing to me. In theorem 4, $ \eta $ is a function of $ \theta$, and $ \theta$ is a random variable distributed according to $ p $. But then $ q_{t, H} (\eta ) $ is a density of $ \eta_t$, and then Eqn (11) contains an average over $ \eta_t$, and no term except $ k_\psi$ has a $ \eta$ term. I'm still not sure I understand Eqn 11 completely. This issue gets even worse in Eqns (12) and (13), where a lot of terms are functions of $ t $ although the variables do not include $ t $.

1. The limitations of this work should be addressed a little bit -- from what I understand, you need to be able to compute kernel functions quickly, and require closed form solutions for $ \nabla \psi $ and $ \nabla \psi^* $.

Minor comments:
1. Above eqn (2), the equality should be $ \theta_{d+1} = \sum_{i=1}^d \theta_i$.

1. The authors keep saying they want to generalize to non-Euclidean geometries. If SVGD considers the geometry induced by the KL-divergence, then isn't this already non-Euclidean?

**Summary Of The Paper:**

The paper introduces algorithms for sampling constrained distributions. Assuming the existence of a mirror map that maps from the primal space to the dual space, the authors introduce an algorithm MSVGD that can perform gradient descent in the dual space, and map back to the primal space. They also propose SVMD, which uses a different algorithm to perform gradient descent in the primal space itself. The authors experimentally validate their algorithms on simulated data and one non-simulated dataset (Fig 4b).



**Summary Of The Review:**

I think the result is intuitive and requires advanced techniques to construct the infinitesimal generator used for constrained sampling.

---

> ### Author Response · Authors · 2021-11-22
> **Response**
>
> Thank you for the time you’ve taken to review our work and for the positive and constructive feedback!  We are glad that you found the premise of our paper clear and interesting and the contributions significant.  We respond to each of your remaining comments below.
>
> ###  Writing
> Following your suggestion, in the latest draft we have removed inessential details from the paragraph between Eqns. (4) and (5) and deferred a more detailed discussion of Stein operators to Appendix C.  In addition, we provide the details of the Section 5.2 experiments in Appendix F.2.  Please let us know if you have any additional suggestions for improving clarity or accessibility.
>
> ### Notation
> We apologize for the notation confusion. We have fixed the notations in eqs. (11) to (13). Please let us know if it resolves the questions you have. You are right that the eigenvalues and eigenfunctions should depend on t in eq. (12) and eq. (13).
>
> ### Limitations
> Your understanding is correct, and we have revised the discussion section to discuss limitations.
>
> ### About $\\theta_{d+1}$
> It should be $1 - \\sum_{i=1}^d$ since $\\sum_{i=1}^{d+1} \\theta_i = 1$ so that the vector $(\\theta_1, \\cdots, \\theta_{d+1})$ is a normalized probability parameter for $d+1$ categories.
>
> ### Non-Euclidean geometry
> We apologize for the confusion: by “non-Euclidean geometry” we mean Riemannian geometry in **the space of the particles** rather than in the space of probability distributions.  SVGD uses Euclidean geometry in particle space in the sense that both MSVGD and SVMD reduce to SVGD if $\\psi(\\theta) = \\frac{1}{2}\\|\theta\\|^2$. In this case $\\nabla\\psi$ becomes the identity map, $\\nabla^2\\psi$ is the identity matrix. And the kernel for SVMD becomes
> $K_{\\psi, t}(\\theta, \\theta’) = E_{\\theta_t\\sim q_t} [k^{1/2}(\\theta, \\theta_t) k^{1/2}(\\theta_t, \\theta’)] I = \\sum_i \\sum_j \\lambda_i^{1/2}\\lambda_j^{1/2} u_i(\\theta)u_j(\\theta’) E_{\\theta_t \\sim q_t} [u_i(\\theta_t) u_j(\\theta_t)] I = \\sum_i \\sum_j \\lambda_i^{1/2}\\lambda_j^{1/2} u_i(\\theta)u_j(\\theta’) \\delta_{i, j} I = k(\\theta, \\theta’) I$.

---

### Decision · Program_Chairs · 2022-01-20

**Decision:**

Accept (Spotlight)

**Comment:**

The paper proposes to extend mirror descent to sampling with stein operator when the density is defined on a constrained domain and non euclidean geometry. All reviewers agreed on the novelty and the merits of the paper. Accept